# AgentSquare: Automatic LLM Agent Search in Modular Design Space

**Yu Shang**[1]*, **Yu Li**[2]*, **Keyu Zhao**[1], **Likai Ma**[1], **Jiahe Liu**[1], **Fengli Xu**[1]†, **Yong Li**[1]†

[1]Department of Electronic Engineering, Tsinghua University
[2]Shenzhen International Graduate School, Tsinghua University
`{fenglixu,liyong07}@tsinghua.edu.cn`

## Abstract

Recent advancements in Large Language Models (LLMs) have led to a rapid growth of agentic systems capable of handling a wide range of complex tasks. However, current research largely relies on manual, task-specific design, limiting their adaptability to novel tasks. In this paper, we introduce a new research problem: Modularized LLM Agent Search (MoLAS). We propose a modular design space that abstracts existing LLM agent designs into four fundamental modules with uniform IO interface: *Planning*, *Reasoning*, *Tool Use*, and *Memory*. Building on this design space, we present a novel LLM agent search framework called AgentSquare, which introduces two core mechanisms, *i.e.*, *module evolution* and *recombination*, to efficiently search for optimized LLM agents. To further accelerate the process, we design a performance predictor that uses in-context surrogate models to skip unpromising agent designs. Extensive experiments across six benchmarks, covering the diverse scenarios of web, embodied, tool use and game applications, show that AgentSquare substantially outperforms hand-crafted agents, achieving an average performance gain of 17.2% against best-known human designs. Moreover, AgentSquare can generate interpretable design insights, enabling a deeper understanding of agentic architecture and its impact on task performance. We believe that the modular design space and AgentSquare search framework offer a platform for fully exploiting the potential of prior successful designs and consolidate the collective efforts of research community. Code repo is available at https://github.com/tsinghua-fib-lab/AgentSquare.

## 1 Introduction

The past few years have witnessed remarkable progress in the development of Large Language Models (LLMs) (Achiam et al., 2023; Touvron et al., 2023), giving rise to the proliferation of numerous agentic systems (Weng, 2023; Shen et al., 2024). For example, "chain-of-thought" prompting has unlocked the general-purpose reasoning capabilities of LLMs (Wei et al., 2022), and memory mechanisms have been proven effective in simulating human behavioiur (Park et al., 2023). These emerging LLM agents have demonstrated astonishing abilities to transform a wide range of tasks, including solving mathematical problems (Romera-Paredes et al., 2024), navigating the web (Nakano et al., 2021), providing financial advice (Ding et al., 2024a) and informing medical decisions (Li et al., 2024a). Therefore, the design of agentic systems plays a crucial role in harnessing the power of LLMs for various downstream applications.

However, current research predominantly relies on manually designed agentic systems tailored for specific tasks, which often depend heavily on expert insight and intensive human labor. Furthermore, these task-specific agent designs frequently struggle to adapt to novel tasks. A few recent studies have explored using LLMs to rewrite and optimize the prompts of existing agents (Fernando et al.,

---

*Equal contribution.
†Corresponding author.

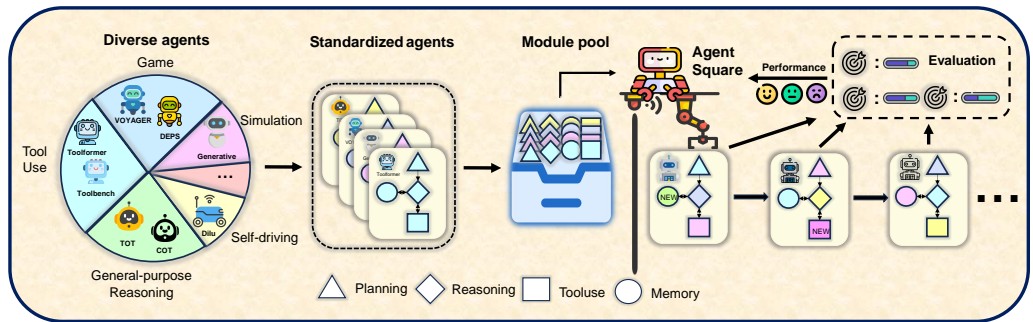

Figure 1: AgentSquare is a modular framework for designing and optimizing LLM agents.

2024; Yang et al., 2024). A more recent work introduces the idea to leverage LLMs to search the entire agentic systems defined in code space (Hu et al., 2024), enabling the discovery of agents with more flexible prompts, control flows, etc. However, these previous approaches are limited in their ability to explicitly recombine the strengths of agentic modules discovered by different researches and located in separate codebases. Another line of research focuses on optimizing the configuration of multi-agent systems (Chen et al., 2023; Yuan et al., 2024; Li et al., 2023; Zhuge et al., 2024; Wang et al., 2023b). These efforts are orthogonal to the optimization of single-agent systems, as they focus more on the role-playing and interaction patterns among multiple agents, rather than the design of agentic modules.

This paper addresses a novel research problem — Modularized LLM Agent Search (MoLAS). The goal is to automatically optimize LLM agent designs by leveraging the experience of published or evaluated modules. Therefore, the core of our work is a modular design space for LLM agents, comprising 4 categories of modules: *Planning*, *Reasoning*, *Tool Use*, and *Memory*. This design space is abstracted from a thorough literature review of existing agentic systems (details provided in Section 2). It is important to note that our goal is not to propose the most comprehensive, one-size-fits-all LLM agent design space, but rather to demonstrate that our modular design space enables researchers and intelligent search algorithms to fully exploit the potential of prior successful designs. MoLAS is a guided and constrained searching problem in the modular design space, which is a subset of the entire code search proposed in ADAS (Hu et al., 2024). However, MoLAS has a nice feature of providing standardized IO interfaces for agent modules, facilitating easy recombination of modules from different agentic systems and hence enabling efficient search for novel agents. Our design space is also highly extensible, allowing new agentic systems to be integrated as plug-in modules. Therefore, it provides a platform to consolidate the collective efforts of the research community on LLM agents. The overview of this work is illustrated in Figure 1.

Building on this modular design space, we propose a novel LLM agent search framework called AgentSquare. Specifically, AgentSquare optimizes LLM agents through the mechanisms of *module evolution* and *recombination*. The *module evolution* mechanism leverages an evolutionary meta-prompt to explore new modules through prompt-level optimization, which jointly models task descriptions, existing modules, and the performance of evaluated modules. Besides, the *module recombination* mechanism performs module-level optimization by leveraging the reasoning power of LLMs to strategically search for promising module combinations. To reduce the expensive evaluation costs of LLM agents, we further introduce a performance predictor that implements an in-context surrogate model for newly proposed LLM agents, enabling us to skip unpromising candidates and significantly accelerate the search process.

We conduct comprehensive evaluations on six widely adopted benchmarks, covering diverse use cases in web, embodied, tool use and game scenarios. Our experiments show AgentSqaure can discover novel LLM agents that outperform hand-crafted agents across all six benchmarks, scoring an average performance gain of 17.2% compared to the best known human designs. Besides, AgentSqaure also surpasses other search algorithms in terms of having a steeper optimization trajectory. More importantly, case studies reveal that AgentSquare can provide human interpretable design insights for newly discovered, good-performing agents.

The key contributions of this work are as follows:

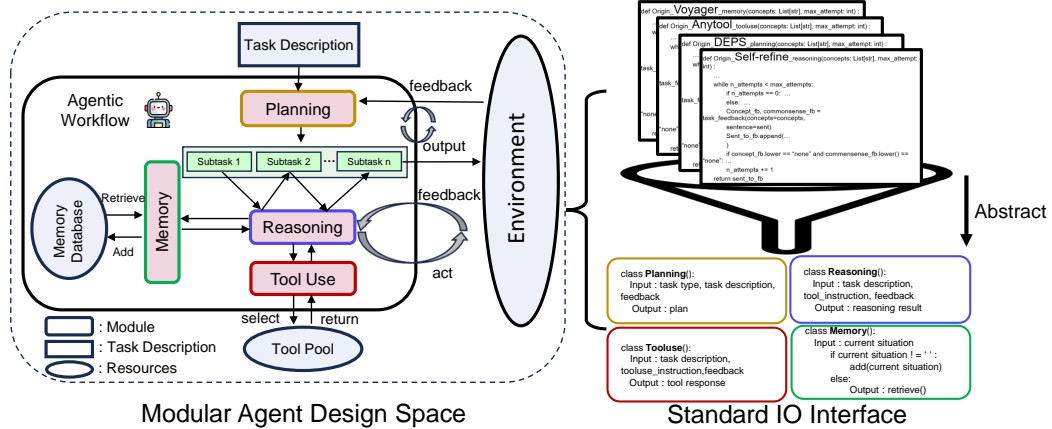

Figure 2: Illustration of the modular agent design space and agentic workflow (left) and the standardized IO interface of four types of modules (right).

- We propose a novel modular design space for LLM agents, enabling researchers to easily build on previous successful designs and accumulate new discoveries as a community.

- We design the AgentSquare framework that efficiently searches for novel and good-performing LLM agents via the novel mechanism of *module evolution*, *module recombination*, and *performance predictor*.

- Experiments across six diverse tasks show that our method discovers novel LLM agents that outperform all known human designs. Besides, AgentSqaure can generate human interpretable design insights for these novel agents.

## 2 A MODULAR DESIGN SPACE OF LLM AGENTS

### 2.1 BACKGROUND

Using LLMs for automatic optimization has been a widely explored topic, such as applications in code generation (Lehman et al., 2023; Romera-Paredes et al., 2024) and neural architecture search (Nasir et al., 2024; Chen et al., 2024a). There are several recent studies that explore the problem of prompting LLMs to design LLM agentic systems. OPRO (Yang et al., 2024) and Prompt-breeder (Fernando et al., 2024) can be viewed as leveraging the reasoning power of LLMs to improve the prompt of LLM agents. More importantly, ADAS introduces the idea of searching the entire agentic system defined in code space, and propose a Meta Agent Search algorithm that discovers LLM agents outperforming state-of-the-art human designs (Hu et al., 2024). Our main difference and contribution lie in introducing a modular design space for LLM agents, which can provide a standard framework to support the convenient reuse of existing successful agent components and fruitful innovative agent module discovery.

A modular design space for LLM agents facilitates the reuse of prior successful designs and supports the exploration of new architectures. At the core of such modularization is the standardization of input-output interfaces, which ensures both extensibility and seamless integration with existing designs. Many experts in the field have proposed building LLM agentic systems with key modular components from engineering (Weng, 2023) and cognitive perspectives (Sumers et al., 2023). However, these proposals remain largely conceptual, lacking implementable solutions to unify existing LLM agents. Besides, current LLM workflow program frameworks (*e.g.*, LangChain and Auto-GPT) only provide operation-level components, which cannot support module-level search that best exploits the potential of prior successful designs.

To address these problems, we perform a comprehensive literature review of publications from NeurIPS, ICML, and ICLR over the past three years. The review focuses on papers with the keywords "LLM", "Agent", or "Large Language Model" in their titles while excluding works related to multi-agent systems or agents that require additional training. Note that our aim is not to propose the most comprehensive, one-for-all LLM agent design space, but to offer a standardized framework

that enables the recombination of existing agents and facilitates the discovery of new ones. As a result, we sort out 16 popular LLM agents and abstract a modular design space with 1050 possible combinations, which can be easily extended when new modules are discovered. Below, we describe the agentic workflow and the function of four modules in our design space.

## 2.2 WORKFLOW OVERVIEW

The proposed agent workflow operates through an iterative process with the interconnection of the above four modules, as shown in Figure 2. Upon receiving a task $d$, the agent starts with the *planning* module, decomposing it into $n$ sub-tasks$\{s_1, s_2, \ldots, s_n\}$. Next, these sub-tasks are passed to the *reasoning* module sequentially. Taking the sub-task $s_i$ description as input, the *reasoning* module explores to prompt LLMs to give the result. When reasoning encounters limitations in internal knowledge of LLMs, the *tool use* module is activated to select an appropriate tool from the pre-defined tool pool $\tau$, supporting problem-solving. Besides, the reasoning process also accesses the *memory* module which reads and writes necessary observations and experiences from a memory database $mem$ to help reasoning. The reasoning result of each sub-task will be transformed into actions, guiding the agent to interact with the external environment. After all sub-tasks are finished or the reasoning process gets stacked, the agent will activate the *planning* module to adjust the plan with the received feedback. The agent conducts such a trial-and-error loop until the task $d$ is completed or the set maximum trial number is reached.

**Planning.** The planning module is responsible for decomposing the targeted task into smaller sub-tasks. Given a task description $d$ and optional feedback information $f$, the planning module $P$ strategically decomposes the targeted task into a sub-task sequence $\{s_1, s_2, \ldots, s_n\} = P(d, f)$. Such decomposition is critical for handling very complex tasks with long-term characteristics, especially for agents in open-world environments such as MineCraft (Wang et al., 2024a;c).

**Reasoning.** LLMs have exhibited remarkable reasoning abilities under advanced prompting approaches such as CoT (Wei et al., 2022), ToT (Yao et al., 2024), and SoT (Shang et al., 2024), shaping the foundation of the intelligence of LLM agents. The reasoning module $R$ is invoked to solve the sub-tasks sequentially after planning, which takes each sub-task $s_i$ and optional feedback information $f_i$ as input and outputs a solution $r_i = R(s_i, f_i)$.

**Tool use.** The ability of using external tools (Shen et al., 2024; Schick et al., 2024) overcomes the limitations of the LLM's internal knowledge during the reasoning process. Formally, given certain problem $p_{ij}$ derived from the reasoning process of sub-task $s_i$ and a pre-defined tool pool $\tau$, the tooluse module $T$ selects the best-matched tool $t_{ij}$ to address the problem, denoted as $t_{ij} = T(p_{ij}, \tau)$, where $t_{ij} \in \tau$.

**Memory.** Memory plays a critical role by storing past thoughts, actions, and observations of agents (Park et al., 2023; Shinn et al., 2024). During the reasoning process, these internal logs are dynamically written to and retrieved from the memory database $mem$, controlled by the memory module $M$. The writing process can be expressed as $mem = M_{write}(o, mem)$, where $o$ denotes the current observations. The retrieval process is $m = M_{retrieve}(o, mem)$, where $m$ denotes the retrieved knowledge relevant to the current situation.

## 3 AGENTSQUARE FRAMEWORK

### 3.1 PROBLEM FORMULATION OF MOLAS

In the proposed modular design space, an LLM agent $A$ can be instantiated with the combination of a planning module $P$, a reasoning module $R$, a tooluse module $T$ and a memory module $M$, denoted as $A = (P, R, T, M)$. Given the task description $d$ and the set of all possible modules with standardized IO interface $\{\mathbb{P}, \mathbb{R}, \mathbb{T}, \mathbb{M}\}$. We formulate an optimization problem for searching LLM agent architectures within the modular design space. The objective is to identify the optimal module combination in a solution space defined by a Cartesian product of four design dimensions to maximize agent performance. Let the performance evaluation function of the task be $Eval_d(\cdot)$, where the specific metric varies in different tasks as discussed in Appendix A.1. The optimization

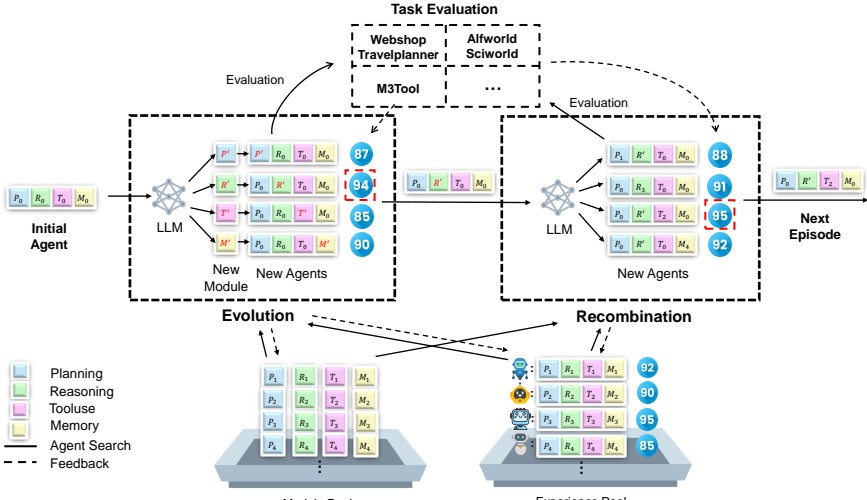

Figure 3: Overview of AgentSquare search framework. AgentSquare optimizes LLM agents through the mechanisms of module evolution and recombination. We further introduce a performance predictor that implements an in- context surrogate model for efficient evaluation of novel agents.

problem of MoLAS is defined as follows:

$$\underset{P \in \mathbb{P}, R \in \mathbb{R}, T \in \mathbb{T}, M \in \mathbb{M}}{\arg\max} Eval_d(P, R, T, M). \tag{1}$$

## 3.2 AGENTSQUARE SEARCH ALGORITHM

Solving the optimization problem of MoLAS features three key challenges: (1) The search space, defined as the Cartesian product of four orthogonal modules, is vast and hard to explore; (2) the module sets encompass any code with standard IO interfaces, making the module selection an open-ended problem; (3) the high costs of agent evaluation during the search process constrain the overall search scale. To tackle these issues, we introduce AgentSquare, an automatic search framework to optimize LLM agents within the modular design space. Facing the vast search space of MoLAS, we propose *module recombination* operation utilizing LLMs to strategically reason to identify more promising module combinations. Such operation broadens the coverage of child samples, overcoming the limitations of prompt rewrite methods that explore only a restricted space. However, only searching in the existing module combinations also narrows the search space, thus we propose *module evolution* operation which employs an evolutionary meta-prompt to search new modules through code-level optimization. This operation, combined with module recombination, enables the search of any module combination in the open-ended solution space. Finally, to mitigate the high costs of frequent evaluations of searched agents, we design a *performance predictor* as an in-context surrogate model for evaluating searched agents, significantly accelerating the search process and reducing real-valued costs.

The overall framework of AgentSquare is illustrated in Figure 3 and the algorithm is presented in Algorithm 1. Next, we detail the key components of the AgentSquare search process.

## 3.3 INITIALIZATION

Insights from existing AutoML studies indicate that a well-chosen initialization enhances warm-up and improves search efficiency by avoiding unpromising populations (So et al., 2019; Yuan et al., 2024). AgentSquare starts by initializing a global experience pool $\mathbb{E} = \{(P, R, T, M, v) | P_0 \in \mathbb{P}, R_0 \in \mathbb{R}, T_0 \in \mathbb{T}, M_0 \in \mathbb{M}\}$ to seed agents that are well-designed (as mentioned in Section 2) along with their real-valued performance $v$. The module pools $\{\mathbb{P}, \mathbb{R}, \mathbb{T}, \mathbb{M}\}$ are set to the standardized modules extracted from these seed agents.

### 3.4 Module Recombination

Given the vast solution space of MoLAS, relying solely on prompt rewriting leads to a limited exploration confined to the neighbor of the initial state. To expand the exploration space, we propose leveraging LLMs as a *self-adaptive proposer*, which iteratively reason to identify promising module combinations with accumulated experience beyond the original agent configuration. Denote the initial agent of the recombination phase as $A_r^0 = (P_0, R_0, T_0, M_0)$, where $P_0 \in \mathbb{P}, R_0 \in \mathbb{R}, T_0 \in \mathbb{T}, M_0 \in \mathbb{M}$. The module combination proposer LLM $\pi_\theta$ incorporates targeted task description $d$, existing module pools $\{\mathbb{P}, \mathbb{R}, \mathbb{T}, \mathbb{M}\}$ and the performance experience of searched module combinations $\mathbb{E}$ to propose promising new agents $A_r$:

$$A_r = \pi_\theta((P_0, R_0, T_0, M_0), d, N, \mathbb{P}, \mathbb{R}, \mathbb{T}, \mathbb{M}, \mathbb{E}). \tag{2}$$

Based on the initial agent configuration $A_r^0$, the LLM proposes $N$ offspring $\{A_r^1, A_r^2, ..., A_r^N\}$ by replacing certain modules of $A_r^0$ with alternatives from the module pool. For instance, a possible solution could be $(P_0, R^{'}, T_0, M_0)$, where $R^{'} \in \mathbb{R}$ is a different reasoning module selected from the module pool. Then, the created $N$ new agents are evaluated with a performance predictor $\pi_p$ (detail in Seciton 3.6) and the best one goes to the next episode as initialization.

### 3.5 Module Evolution

As mentioned above, the solution space for each module type is open-ended, allowing any code with a standardized I/O interface. Consequently, searching only with module recombination narrows the solution space and limits the upper bound of agent performance. To address this problem, we design a *module evolution* operation with an evolutionary meta-prompt to search for new modules through program-level optimization. This design is inspired by the iterative pipeline of FunSearch (Romera-Paredes et al., 2024), which prompts LLMs to propose new solutions based on the target problem and performance feedback from existing solutions. Building on this concept, we introduce a module-programming LLM $\pi_\xi$ to conduct agent search in our modular design space by jointly modeling task descriptions, existing modules, and the performance of previously evaluated modules. Please note we reuse parts of the open-source code from ADAS (Hu et al., 2024) to implement the optimization procedure. Leveraging LLMs to search in the modular agent design space has several appealing advantages. Compared with the unconstrained design space of LLM agents, searching functional modules can produce a more focused and fruitful search space. Additionally, integrating existing successful module designs with standard IO as in-context examples can better elicit the reflective reasoning abilities of LLMs to identify previous key designs to help propose innovative ones. Denote the initial agent in the module evolution stage as $A_e^0 = (P_0^{'}, R_0^{'}, T_0^{'}, M_0^{'})$, the module programmer LLM produces a population of child agents by evolving current modules of $A_e^0$. Formally the module evolution operation is denoted as follows:

$$A_e = \pi_\xi((P_0^{'}, R_0^{'}, T_0^{'}, M_0^{'}), d, N, \mathbb{P}, \mathbb{R}, \mathbb{T}, \mathbb{M}, \mathbb{E}). \tag{3}$$

The created new modules are appended to the standardized module pools $\{\mathbb{P}, \mathbb{R}, \mathbb{T}, \mathbb{M}\}$ and each module is used to individually mutate the initial agent, resulting in $N$ child agents $\{A_e^1, A_e^2, ..., A_e^N\}$. For example, $(P^*, R_0, T_0, M_0)$ represents a solution where the planning module is mutated into a new variant $P^*$. These child agents are then real-tested and updated to the historical experience pool $\mathbb{E}$. The best-performing one is selected as the initial agent for the subsequent recombination phase.

### 3.6 Performance Predictor

The last challenge in automatic agent search is the high API cost incurred during the evaluation of each candidate agent. Many agent tasks require multiple steps and involve substantial input and output tokens, leading to prohibitive evaluation costs. For instance, evaluating a simple CoT agent based on GPT-4o in ALFWorld (Shridhar et al., 2021) requires around $60, making the agent search economically unsustainable at scale. To tackle this issue, we propose incorporating an additional LLM $\pi_p$ as a performance predictor to serve as an in-context surrogate model for novel agent evaluation, enabling the exclusion of unpromising candidates and significantly accelerating the search process. Compared to real environment evaluation, such an in-context surrogate model requires significantly fewer tokens, making it more cost-efficient and supporting larger-scale searches. Similar approaches have been effectively applied in neural architecture search (NAS), where LLMs are

| | | Web | Embodied | | Tool | | Game |
|---|---|---|---|---|---|---|---|
| **Baseline Type** | **Method** | **Webshop** | **ALFWorld** | **SciWorld** | **M3Tool** | **Travel** | **PDDL** |
| | CoT | 0.485 | 0.405 | 0.697 | 0.448 | 0.487 | 0.542 |
| | Cot-SC | 0.512 | 0.426 | 0.656 | 0.461 | 0.413 | 0.495 |
| | Self-refine | 0.461 | 0.567 | 0.654 | 0.442 | 0.000 | 0.514 |
| | ToT | 0.501 | 0.437 | 0.741 | 0.453 | 0.380 | 0.476 |
| | Step Back | 0.468 | 0.279 | 0.220 | 0.434 | 0.000 | 0.486 |
| | TP | 0.398 | 0.404 | 0.576 | 0.387 | 0.430 | 0.518 |
| Hand-crafted Agents | HuggingGPT | 0.519 | 0.481 | 0.680 | 0.354 | 0.510 | 0.584 |
| | Voyager | 0.366 | 0.425 | 0.776 | 0.247 | 0.523 | 0.412 |
| | Generative Agents | 0.499 | 0.477 | 0.663 | 0.402 | 0.480 | 0.553 |
| | DEPS | 0.481 | 0.459 | 0.740 | 0.278 | 0.540 | 0.591 |
| | OPENAGI | 0.506 | 0.510 | 0.718 | 0.322 | 0.533 | 0.616 |
| | Dilu | 0.451 | 0.433 | 0.682 | 0.475 | 0.360 | 0.463 |
| Module Search | Random | 0.533 | 0.620 | 0.704 | 0.438 | 0.563 | 0.660 |
| | Bayesian | 0.549 | 0.634 | 0.749 | 0.502 | 0.537 | 0.650 |
| Prompt Search | OPRO | 0.505 | 0.380 | 0.569 | 0.309 | 0.523 | 0.589 |
| Agent Search | ADAS | 0.521 | 0.543 | 0.754 | 0.475 | 0.373 | 0.568 |
| | **AgentSquare** | **0.607** | **0.695** | **0.781** | **0.524** | **0.583** | **0.669** |

Table 1: Performance comparison of searched agents from AgentSquare and (1) existing human-designed agents (2) module search baselines (3) prompt search baselines (4) agent search baselines based on GPT-4o on six tasks across different domains.

leveraged to evaluate the performance of generated network architectures (Jawahar et al., 2023; Chen et al., 2024a).

During the search process, newly created agents from module evolution are still tested in the real task environment because these new modules never appear in the experience pool, and it is unsuitable to use the performance predictor to provide predictions. During the module recombination operation, the newly proposed agents are evaluated by the performance predictor, which leverages in-context reasoning based on past agent combination performance to provide efficient performance prediction. Here, given a newly searched agent $A'$, the performance predictor $\pi_p$ thoroughly considers task descriptions $d$, module profiles and in-context performance examples of previously tested agents $\mathbb{E}$ to score novel agents:

$$v' = \pi_p(A', d, \mathbb{P}, \mathbb{R}, \mathbb{T}, \mathbb{M}, \mathbb{E}), \tag{4}$$

where $v'$ is the predicted performance of the evaluated agent. Empirical results demonstrate that the predicted performance of agents closely matches their actual performance, verifying the effectiveness of the proposed performance predictor, which is detailed in Section 4.3.

## 4 EXPERIMENTS

### 4.1 EXPERIMENTAL SETUP

**Task setup.** We conduct experiments on six representative tasks covering four domains: embodied, game, web and tool applications, which are widely adopted by existing LLM agent benchmarks (Ma et al., 2024; Xi et al., 2024), more details are presented in Appendix A.1.

**Baselines.** We compare AgentSquare with four types of baselines including hand-crafted agents, module-level search, prompt-level search and agent-search methods. More details are presented in Appendix A.1.

**AgentSquare setup.** We implement AgentSquare and conduct experiments using both GPT-3.5-turbo-0125 and GPT-4o (Achiam et al., 2023). To ensure a fair comparison, we use the same number of few-shot examples across all methods. The initial agent is set as a random module combination, and the search process terminates after 5 consecutive iterations without performance improvement.

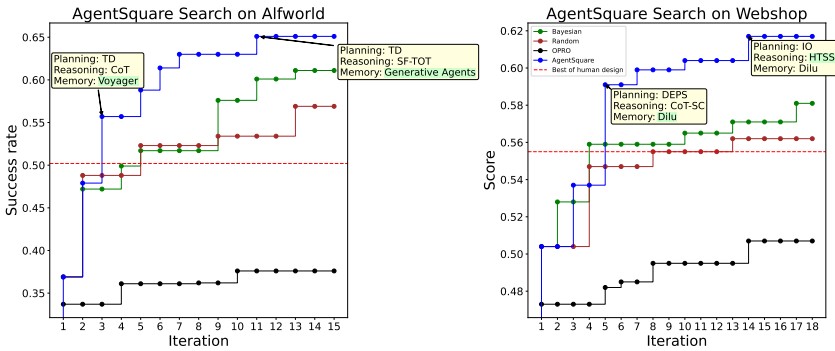

Figure 4: AgentSquare search trajectory on Alfworld and Webhop.

## 4.2 EXPERIMENTAL RESULTS

**Main results.** We conduct extensive experiments to compare our method against three types of baselines on six tasks and present results based on GPT-4o in Table 1 and results on GPT-3.5 in Table A.3. Additionally, we evaluate the agents' API costs and provide a performance-cost comparison in Figure A.7 to Figure A.12. From these results, we have the following observations:

- **AgentSquare can effectively discover better agents compared with human-designed agents.**
  On the six representative agent tasks, the best agent searched by AgentSquare consistently outperforms human-designed agents in terms of performance. Specifically, as shown in Table 1 and Table A.3, compared with the best human-designed agent, AgentSquare achieves an average 14.1% performance improvement on Webshop, 26.1% improvement on ALFWorld, 20.5% improvement on SciWorld, 30.6% improvement on M3Tool, 6.0% improvement on Travelplanner, 6.0% improvement on PDDL. Simultaneously, the best agent from AgentSquare is commonly cost-efficient, which strikes the best performance-cost trade-off among all compared agents as seen in Figure A.7 -Figure A.12. Since the search cost is a one-time expense and the searched modules can be reused, it is not included in the above analysis, but separately listed in Table A.6.

- **AgentSquare provides a more efficient searching approach for LLM agent optimization.** To further demonstrate the effectiveness of the search of AgentSquare, we compare three types of searching methods including module search, prompt search and agent search. Compared with the best agent crafted from these searching methods, AgentSquare achieves an average 8.4% performance improvement on Webshop, 8.1% improvement on ALFWorld, 11.0% improvement on SciWorld, 12.8% improvement on M3Tool, 2.5% improvement on Travelplanner, 1.4% improvement on PDDL. The comparison of search-based methods is conducted with a fixed LLM token budget to ensure fairness by maintaining the same number of search iterations. While in principle ADAS has the potential to discover more sophisticated agents by searching in the entire code space, it may require more iterations (and thus higher LLM token usage) to achieve this.

**Search trajectory in AgentSquare.** We present the search trajectory under 15 iterations using AgentSquare based on GPT-4o and other searching methods on ALFWorld and Webhop tasks in Figure 4. Results on other tasks are presented in Figure A.13 and A.14. AgentSquare demonstrates a steady convergence trajectory, where more advanced agents are continually emerging during search. In contrast, module-level searching methods including random and Bayesian search lack a clear and insightful search direction. Prompt-level search methods such as OPRO are constrained by a limited modification space, leading to minimal performance improvements. As a result, they all encounter performance bottlenecks during the search process, resulting in sub-optimal agent architectures. Besides, we find that simple module-level search methods such as random recombination greatly outperforms prompt-level search, indicating the importance of searching in the modular design space.

## 4.3 ABLATION STUDY OF AGENTSQUARE

**Effectiveness of module evolution and recombination.** There are two key operations in the searching framework of AgentSquare: module evolution which creates new modules and module recom-

| Method | Webshop | ALFWorld | SciWorld | M3Tool | TravelPlanner | PDDL |
|---|---|---|---|---|---|---|
| AgentSquare (full) | **0.607** | **0.695** | **0.781** | **0.524** | **0.583** | **0.669** |
| w/o module evolution | 0.564 | 0.649 | 0.736 | 0.502 | 0.577 | 0.614 |
| w/o module recombination | 0.560 | 0.616 | 0.710 | 0.481 | 0.280 | 0.669 |

Table 2: Ablation study of AgentSquare on GPT-4o on six tasks across different domains.

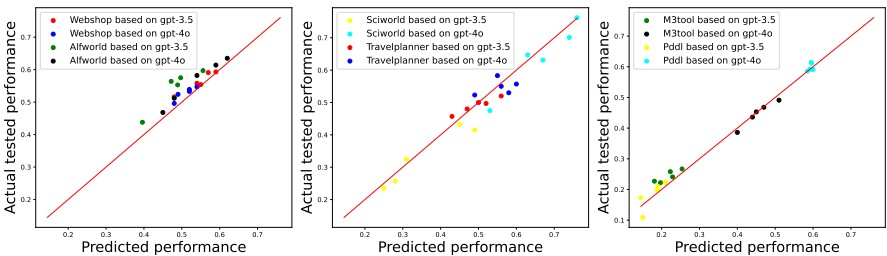

Figure 5: Validation of the effectiveness of the performance predictor (correlation between the actual and predicted performance) on each task.

bination which strategically recombines existing ones. To verify the effectiveness of each design, we tested three variants: the full model, a version without module evolution, and a version without module recombination. The results based on GPT-4o and GPT-3.5 are presented in Table 2 and Table A.5, respectively. It can be seen that dropping each design results in a noticeable performance decline and the module recombination has a larger impact. Module recombination significantly expands the search space, reducing the risk of falling into a local optima. Meanwhile, module evolution facilitates the discovery of more advanced modules tailored to specific tasks. These two operations collaborate well ensuring the effectiveness of the search process in AgentSquare.

**Effectiveness of performance predictor.** In this part, we verify the effectiveness of this design empirically. Figure 5 illustrates the predicted performance of given agents versus their actual tested performance based on both GPT-3.5 and GPT-4o on all six tasks. The tested agents were generated through random sampling by randomly combining existing modules. It can be found that the predicted performance closely aligns with the actual performance, demonstrating the effectiveness of the performance predictor. For instance, the evaluation cost of the predictor is only about 0.025% of the cost of a full evaluation based on GPT-4o in ALFWorld, demonstrating its remarkable cost-efficiency. We provide more experiment results of predicting the performance of the dynamically searched agents in Figure A.15 of the Appendix.

## 4.4 DISCOVERED BEST AGENTS FROM AGENTSQUARE

In this section, we provide some illustrations of the searched best agents, especially some discovered promising modules. Table A.4 summarizes the searched best agent from AgentSquare and the best hand-crafted agents on all tasks. We can observe that AgentSquare can adaptively identify promising agents with both previously existing and newly programmed modules tailored to the given task. For instance, the discovered best agent for ALFWorld combines an existing well-designed memory module from *Generative Agents* with newly created planning (named *TD*) and reasoning modules (named *SF-ToT*). By comparison, the best hand-crafted agent *Self-refine* focuses only on reasoning module design while overlooking other functional modules, leading to suboptimal performance. Moreover, we illustrate two new modules and the human interpretable design insights discovered on ALFWorld in Figure 6. More illustrations are listed in the Figure A.16 to Figure A.21.

## 5 RELATED WORK

### 5.1 LLM-BASED AUTONOMOUS AGENTS

LLM-based autonomous agents are an advanced AI system using a core LLM to manage external functional modules and interact with the world (Ding et al., 2024b). Recent studies have equipped

Figure 6: New module discovered through AgentSquare search on ALFWorld.

LLM agents with several LLM-centric functional modules including planning (Hao et al., 2023; Zeng et al., 2024; Shao et al., 2025), reasoning (Wei et al., 2022; Yao et al., 2024; Shang et al., 2024; Xu et al., 2025), using tools (Shen et al., 2024; Schick et al., 2024), and monitoring memory (Wang et al., 2024a; Park et al., 2023), greatly enhancing the capabilities of LLM agents. Along with the improvement of the single agent, there's another line of work trying to build more advanced multi-agent systems by strategically organizing individual agents for both simulation (Li et al., 2023; Chen et al., 2023) and targeted task solving (Qian et al., 2024; Chen et al., 2024b; Li et al., 2024b). The emergence of more and more sophisticated agent produces remarkable performance improvement, however, their architectures and codebases differ greatly with each other. The lack of a unified design space and consistent terminologies across individual works makes it hard to compare different agents, understand their evolution routes, and guide new agent design directions.

## 5.2 Automatic Design of LLM-based Agents

LLM-based agent system, as the most advanced AI system, has not yet formed a unified design space and an automatic design approach. Engineering-oriented open resources like LangChain[*] and BabyAGI[†] have provided convenient ways to build an LLM-centric agentic system, however, they still need human participation to organize different modules and can't support the optimization of the designed agent. Besides, there have been some conceptual frameworks trying to provide a unified design principle of LLM agents, such as CoALA (Sumers et al., 2023). However, it's still a vision of how LLM agents should be in the future, without providing a practical design framework. More importantly, there are several recent works that explore the problem of automating (at least part of) the design of LLM agent systems defined on different search spaces. OPRO (Yang et al., 2024) and Promptbreeder (Fernando et al., 2024) can be considered as using LLMs to optimize LLM agent defined on prompt space. More relevantly, ADAS (Hu et al., 2024) proposes to search the entire agentic system defined on code space, enabling the search for LLM agents with more flexible prompts, tool uses, control flows and more.

## 6 Conclusion

In this work, we introduce a novel modular design space for LLM agents, allowing researchers to build upon successful prior designs and collectively accumulate new insights. Based on this, we propose a novel research problem, Modularized LLM Agent Search (MoLAS), which aims to automatically optimize LLM agent designs by leveraging the knowledge gained from previously published or evaluated modules. To address the challenge of vast search spaces, we present AgentSquare, an automatic search framework to optimize LLM agents through module evolution and recombination. We further introduce a performance predictor as an in-context surrogate model for evaluating novel LLM agents to accelerate the search process. Overall, our work offers a transition from studying individual LLM agent designs to studying LLM agents within a modular design space, further consolidating the collective efforts of the research community.

---

[*]https://github.com/langchain-ai/langchain
[†]https://github.com/yoheinakajima/babyagi

ACKNOWLEDGMENTS

This work is supported by the National Natural Science Foundation of China under 23IAA02114 and 62472241.

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

# A  APPENDIX

## A.1  EXPERIMENTAL SETUP

**Task setup.** We evaluate AgentSquare and compared methods on six representative tasks covering four key domains which are widely adopted by existing LLM agent benchmarks (Ma et al., 2024; Xi et al., 2024):

- **Embodied:** ALFWorld (Shridhar et al., 2021) with text-based household tasks where agents navigate and interact with objects using text commands, ScienceWorld (Wang et al., 2022) with interactive science tasks requiring agents to navigate rooms and perform experiments, testing scientific commonsense;
- **Game:** PDDL (Ma et al., 2024) including many strategic games where agents use PDDL expressions to complete tasks;
- **Web:** WebShop (Yao et al., 2022) focusing on online shopping tasks where agents browse and purchase products based on user instructions;
- **Tool:** TravelPlanner (Xie et al., 2024) with many travel planning tasks where agents use tools and data to create detailed plans, (6)M3ToolEval (Wang et al., 2024b) including complex tasks requiring multi-turn interactions with multiple tools.

The specific performance evaluation metric varies in different tasks, following the evaluation settings in their original work. Specifically, the evaluation metric is "success rate" for ALFWorld and M3ToolEval, "task score (defined as the average reward obtained across episodes)" for Webshop, "progress rate" for SciWorld and PDDL, and "micro pass rate" for TravelPlanner.

**Baselines.** We compare AgentSquare with four types of baselines:

- **Hand-crafted agents.** We compare with 12 hand-crafted agents including CoT (Wei et al., 2022), CoT-SC (Wang et al., 2023a), Self-refine (Madaan et al., 2024), ToT (Yao et al., 2024), Step back (Zheng et al., 2024), Thought propagation (Yu et al., 2024), HuggingGPT (Shen et al., 2024), Voyager (Wang et al., 2024a), Generative Agents (Park et al., 2023), DEPS (Wang et al., 2024c), OPENAGI (Ge et al., 2024)and Dilu (Wen et al., 2024).
- **Module search methods.** We compare with two module-level agent optimization methods including the random combination of existing modules and Bayesian (Zhou et al., 2019) module combination optimization inspired by Bayesian optimization in NAS (White et al., 2021).
- **Prompt search methods.** We select OPRO (Yang et al., 2024) as a representative prompt-level optimization approach, which leverages LLMs as optimizers by generating and refining instructions through iterative prompts.
- **Agent search methods.** We select ADAS (Hu et al., 2024) which optimizes the entire agentic system in code space as the agent search baseline. We use the official code of ADAS and make slight modifications to adapt it to our tasks.

**AgentSquare setup.** We implement AgentSquare and conduct experiments using both GPT-3.5-turbo-0125 and GPT-4o (Achiam et al., 2023). To ensure a fair comparison, we use the same number of few-shot examples across all methods. The initial agent is set as a random module combination, and the search process terminates after 5 consecutive iterations without performance improvement.

**Algorithm 1:** Algorithm of AgentSquare

**Input:** Initial agent $A_0$, targeted task descriptions $d$, maximum evolution episode $K$,
population size $N$ per evolution phase, standardized module pools $\{\mathbb{P}, \mathbb{R}, \mathbb{T}, \mathbb{M}\}$,
experience pool $\mathbb{E}$
**Output:** The evolved agent $A^*$
$t \leftarrow 1$ // Current search episode
$A_e^0 \leftarrow A_0$ // Initialization of the module evolution phase
**while** $t \leq K$ **do**
$\quad \{A_e^1, A_e^2, ..., A_e^N\} \leftarrow \pi_\xi(A_e^0, d, N, \mathbb{P}, \mathbb{R}, \mathbb{T}, \mathbb{M}, \mathbb{E})$ // Module evolution
$\quad A_r^0 \leftarrow \arg\max\{Eval_d(A_e^0), Eval_d(A_e^1), ..., Eval_d(A_e^N)\}$ // Select the
$\quad\quad$ best-performing generated agent
$\quad \{A_r^1, A_r^2, ..., A_r^N\} \leftarrow \pi_\theta(A_r^0, d, N, \mathbb{P}, \mathbb{R}, \mathbb{T}, \mathbb{M}, \mathbb{E})$ // Module recombination
$\quad A_e^0 \leftarrow \arg\max\{Eval_d(A_r^0), Eval_d(A_r^1), ..., Eval_d(A_r^N)\}$ // Select the
$\quad\quad$ best-performing generated agent
$\quad t \leftarrow t + 1$
**end**
$A^* \leftarrow A_e^0$
**return** $A^*$

| Method Type | Method | Web | Embodied | | Tool | | Game |
| | | Webshop | ALFWorld | SciWorld | M3Tool | Travel | PDDL |
|---|---|---|---|---|---|---|---|
| | CoT | 0.504 | 0.369 | 0.142 | 0.172 | 0.080 | 0.151 |
| | CoT-SC | 0.527 | 0.381 | 0.105 | 0.181 | 0.167 | 0.178 |
| | Self-refine | 0.439 | 0.388 | 0.222 | 0.098 | 0.000 | 0.109 |
| | ToT | 0.510 | 0.381 | 0.143 | 0.189 | 0.163 | 0.147 |
| | Step Back | 0.478 | 0.375 | 0.027 | 0.128 | 0.120 | 0.137 |
| | TP | 0.429 | 0.299 | 0.168 | 0.139 | 0.063 | 0.122 |
| Hand-crafted Agents | HuggingGPT | 0.518 | 0.502 | 0.270 | 0.012 | 0.470 | 0.212 |
| | Voyager | 0.427 | 0.369 | 0.301 | 0.008 | 0.480 | 0.149 |
| | Generative Agents | 0.539 | 0.388 | 0.153 | 0.144 | 0.060 | 0.123 |
| | DEPS | 0.555 | 0.474 | 0.308 | 0.017 | 0.500 | 0.186 |
| | OPENAGI | 0.507 | 0.448 | 0.257 | 0.008 | 0.430 | 0.178 |
| | Dilu | 0.418 | 0.291 | 0.000 | 0.131 | 0.137 | 0.054 |
| Module Search | Random | 0.562 | 0.569 | 0.367 | 0.235 | 0.473 | 0.216 |
| | Bayesian | 0.581 | 0.611 | 0.269 | 0.217 | 0.497 | 0.210 |
| Prompt Search | OPRO | 0.507 | 0.376 | 0.032 | 0.193 | 0.513 | 0.179 |
| Agent Search | ADAS | 0.519 | 0.274 | 0.217 | 0.193 | 0.410 | 0.186 |
| | **AgentSquare** | **0.617** | **0.651** | **0.432** | **0.285** | **0.520** | **0.219** |

Table A.3: Performance comparison of searched agents from AgentSquare and (1) existing human-designed agents (2) module search baselines (3) prompt search baselines based on GPT-3.5 on six tasks across different domains.

| Task | Planning | Reasoning | Tooluse | Memory | Best Hand-crafted Agents |
|---|---|---|---|---|---|
| Webshop | IO | HTSS | / | Dilu | HuggingGPT |
| ALFWorld | TD | SF-ToT | / | Generative Agents | Self-refine |
| SciWorld | Voyager | CoT | / | Hier | Voyager |
| M3Tool | / | CoT-SC | ToolBF | / | Toolbench |
| TravelPlanner | DEPS | CoT | TH | / | DEPS |
| PDDL | IR | CASRC | / | Generative Agents | OPENAGI |

Table A.4: Comparison between the searched best agent from AgentSquare and the best human-designed agent on all tasks.

| Method | Webshop | ALFWorld | SciWorld | M3Tool | TravelPlanner | PDDL |
|---|---|---|---|---|---|---|
| AgentSquare(full) | **0.617** | **0.651** | **0.432** | **0.285** | **0.520** | **0.219** |
| w/o module evolution | 0.595 | 0.623 | 0.288 | 0.236 | 0.483 | 0.202 |
| w/o module recombination | 0.578 | 0.546 | 0.310 | 0.258 | 0.267 | 0.173 |

Table A.5: Ablation study of AgentSquare on GPT-3.5 on six tasks across different domains.

| | Webshop | ALFWorld | SciWorld | M3Tool | TravelPlanner | PDDL |
|---|---|---|---|---|---|---|
| Avg cost (GPT-3.5) | $3.16 | $4.25 | $1.92 | $2.43 | $1.84 | $2.70 |
| Iterations (GPT-3.5) | 23 | 21 | 8 | 14 | 9 | 17 |
| Avg cost (GPT-4o) | $10.51 | $13.96 | $42.14 | $26.03 | $29.75 | $26.94 |
| Iterations (GPT-4o) | 18 | 15 | 9 | 18 | 8 | 12 |

Table A.6: Average API cost per search iteration and the total number of iterations until termination for AgentSquare using GPT-3.5 and GPT-4o across six tasks.

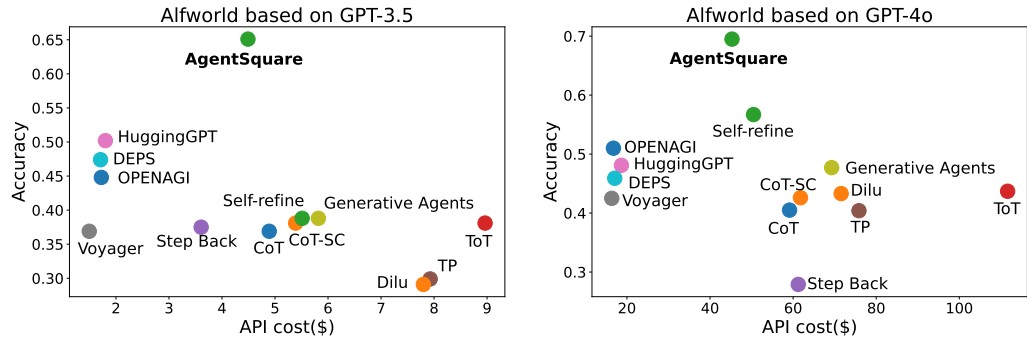

Figure A.7: Performance versus API costs visualization on ALFWorld task.

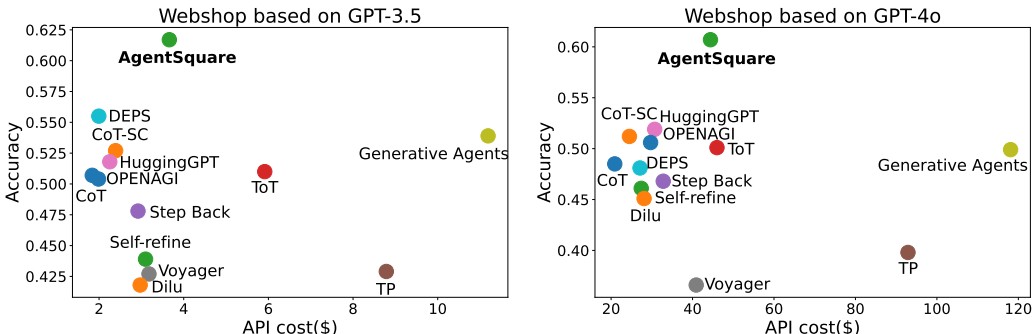

Figure A.8: Performance versus API costs visualization on Webshop.

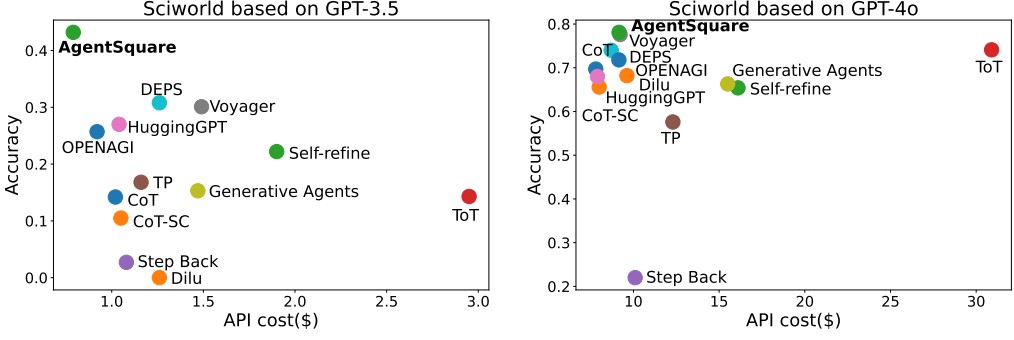

Figure A.9: Performance versus API costs visualization on Sciworld.

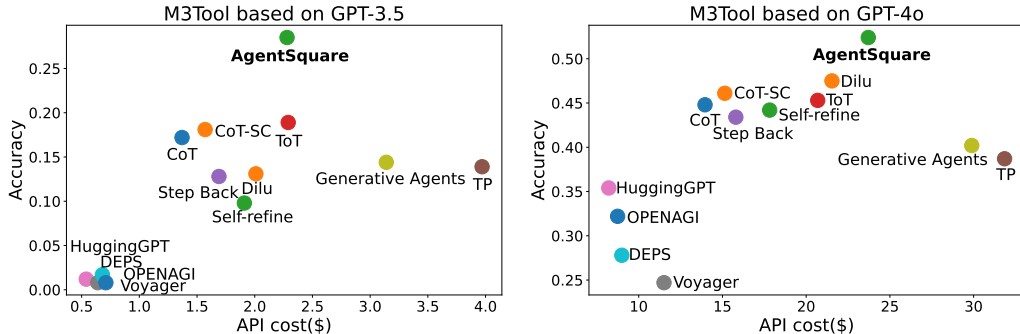

Figure A.10: Performance versus API costs visualization on M3tool.

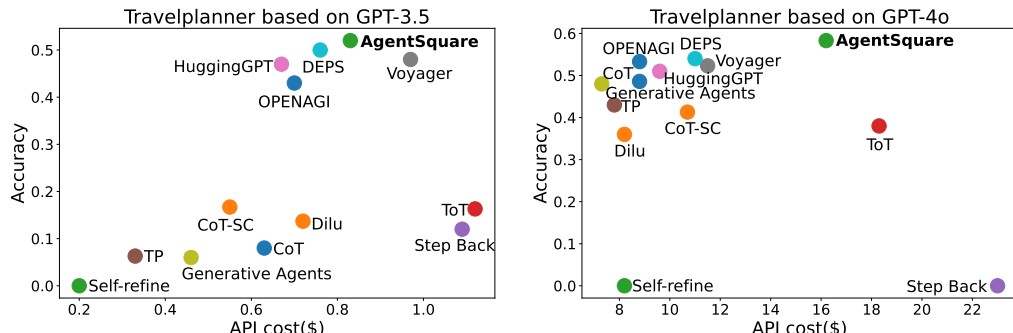

Figure A.11: Performance versus API costs visualization on Travelplanner.

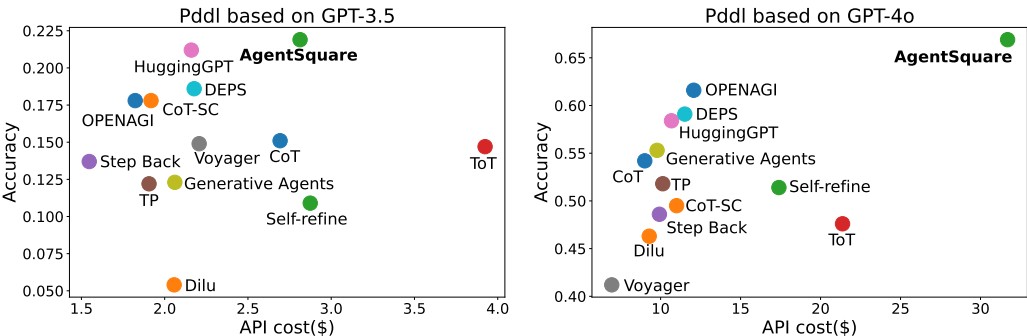

Figure A.12: Performance versus API costs visualization on PDDL.

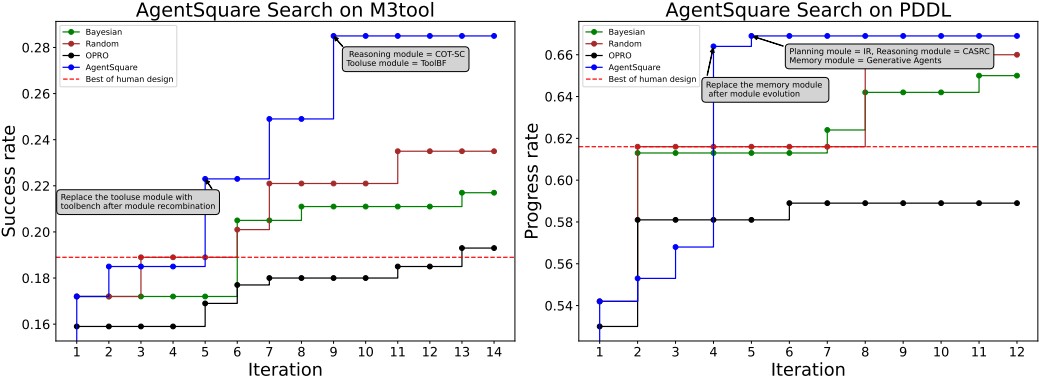

Figure A.13: AgentSquare search trajectory on M3tool and PDDL (more hand-crafted agents, specific module combinations when surpassing best hand-crafted and the final evolved agent, other search baselines).

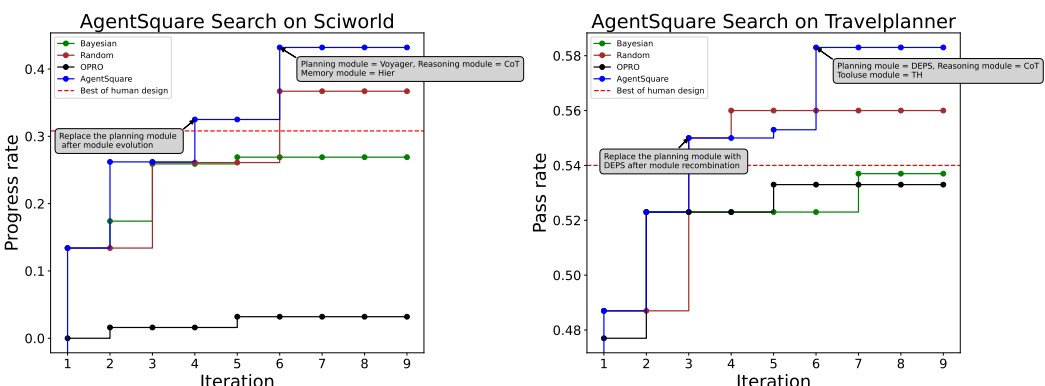

Figure A.14: AgentSquare search trajectory on Sciworld and Travelplanner (more hand-crafted agents, specific module combinations when surpassing best hand-crafted and the final evolved agent, other search baselines).

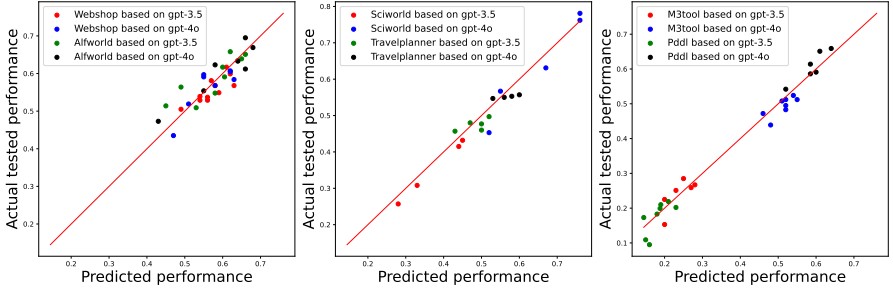

Figure A.15: Validation of the effectiveness of the performance predictor on dynamically searched agents for each task.

**HTSS**

**Insights:** Observing the current performance of the reasoning modules, it seems that techniques like Chain-of-Thought (CoT) and Tree-of-Thoughts (ToT) have offered improvements by breaking down the task into smaller steps and evaluating multiple reasoning paths. The Self-Consistency (SC) approach also shows promise by generating multiple answers and voting on them, while the Self-Refine module uses iterative improvement based on feedback.
**Overall Idea:** To further enhance the performance, we can integrate a combination of these techniques into a single module. Specifically, we can combine the Tree-of-Thoughts (ToT) approach with Self-Consistency (SC) and Self-Refine. This combined approach would involve generating multiple reasoning paths, evaluating them to choose the best path, and then iteratively refining the chosen path based on feedback.
**Implementation:** 1. Generate Multiple Reasoning Paths: Use the Tree-of-Thoughts (ToT) approach to generate multiple reasoning paths.2. Evaluate and Choose the Best Path: Use Self-Consistency (SC) to evaluate these paths by voting and selecting the most common or accurate path.3. Refine the Chosen Path: Use Self-Refine to iteratively improve the chosen path based on feedback.4. Prompt Structure: Craft prompts that encourage step-by-step problem-solving, refer to similar solved examples, and refine output based on feedback.

```python
class REASONING_HYBRID_TOT_SC_SELFREFINE():
    def __init__(self, profile_type_prompt, memory, tooluse, llms_type):
        self.feedback = ''
        self.profile_type_prompt = profile_type_prompt
        self.memory = memory
        self.llm_type = llms_type[0]
        self.tooluse = tooluse
    def __call__(self, task_description: str, tool_instruction :str='',
         feedback :str= ''):
        task_name = re.findall(r'Instruction:\s+(.*?)\s+\[Search\]',
            task_description)
        if self.memory is not None:
            self.task_name_cache = task_name[1]
            self.memory_cache = self.memory(task_description)
            if task_description.count('Reasoning') == 2:
                self.memory_cache = self.memory_cache.split('Observation')[0]
            elif task_description.count('Reasoning') == 4:
                self.memory_cache =
                    'Observation'.join(self.memory_cache.split('Observation')[0:3])
            else:
                self.memory_cache = self.memory_cache
        else:
            self.memory_cache = ''
        if self.tooluse is not None:
            tooluse = self.tooluse(task_description, tool_instruction)
        else:
            tooluse = ''
        split_text = task_description.rsplit('WebShop', 1)
        examples = split_text[0]
        task_description = 'WebShop' + split_text[1]
        prompt = '''{tooluse}Solve the task step by step. Your output must
            follow the examples process. Don't refine your search. You have
            to choose one from a list of items.
{memory}{examples}
{task_description}'''
        prompt = prompt.format(task_description=task_description,
            examples=examples, memory=self.memory_cache, tooluse=tooluse)
        reasoning_results = llm_response(prompt=prompt, model=self.llm_type,
            temperature=0.1, stop_strs=['\n'], n=3)
        # Evaluate and choose the best path
        from collections import Counter
        string_counts = Counter(reasoning_results)
        best_path = string_counts.most_common(1)[0][0]
        # Refine the chosen path based on feedback
        refined_result = self.refine(best_path)
        #reasoning_result = self.refine(reasoning_result)
        return refined_result
```

Figure A.16: New module discovered through AgentSquare search on Webshop.

---

**Hier**

**Insights:** The proposed memory module's hierarchical structure offers significant advantages in task management for intelligent agents. By breaking down each task into smaller sub-tasks stored separately, the system enables focused information retrieval, allowing the agent to access only the relevant data instead of sifting through an entire task trajectory.
**Overall Idea:** My proposed memory module will focus on creating a hierarchical memory structure, where each task is broken down into smaller sub-tasks and each sub-task is stored separately. This approach allows the agent to retrieve focused information on specific sub-tasks rather than an entire task trajectory. Additionally, this memory module will include a feedback mechanism to improve memory relevance and accuracy over time.
**Implementation:** The implementation involves modifying the memory module to store and retrieve sub-task trajectories and introduce a feedback loop for continuous improvement.

```python
class MEMORY_HIER():
    def __init__(self, llms_type) -> None:
        self.llm_type = llms_type[0]
        self.embedding = OpenAIEmbeddings()
        db_path = os.path.join('./db', 'memory/')
        if os.path.exists(db_path):
            shutil.rmtree(db_path)
        self.scenario_memory = Chroma(
            embedding_function=self.embedding,
            persist_directory=db_path
        )
    def __call__(self, current_situation: str=''):
        if 'The correct trajectory is' in current_situation:
            self.addMemory(current_situation)
        else:
            return self.retrieveMemory(current_situation)

    def retrieveMemory(self, query_scenario):
        sub_task_name = query_scenario
        if self.scenario_memory._collection.count() == 0:
            print("The memory vector database is empty. Cannot perform
                search.")
            return ''
        similarity_results = self.scenario_memory.similarity_search_with_score(
            sub_task_name, k=3)
        fewshot_results = []
        for idx in range(0, len(similarity_results)):
            fewshot_results.append(similarity_results[idx][0].metadata
            ['sub_task_trajectory'])
        return "\nHere are similar sub-tasks and the correct handling
            trajectories in these cases: " + ', '.join(fewshot_results)
    def addMemory(self, current_situation):
        sub_task_trajectory = current_situation
        sub_task_name = re.search(r'Sub-task : (.*?) \nThe correct trajectory
            is', current_situation)
        if sub_task_name is not None:
            sub_task_descrip = sub_task_name.group(1)
            doc = Document(
                page_content=sub_task_descrip,
                metadata={"sub_task_name": sub_task_descrip,
                        'sub_task_trajectory': sub_task_trajectory}
            )
            id = self.scenario_memory.add_documents([doc])
    def feedbackMechanism(self, current_situation, was_successful):
        if was_successful:
            self.addMemory(current_situation)
        else:
            sub_task_name = re.search(r'Sub-task : (.*?) \nThe incorrect
                trajectory is', current_situation)
            if sub_task_name is not None:
                sub_task_descrip = sub_task_name.group(1)
                doc_id =
                    self.scenario_memory.search_documents(sub_task_descrip)[0].id
                self.scenario_memory.delete_document(doc_id)
```

Figure A.17: New module discovered through AgentSquare search on Sciworld.

---

**ToolBF**

**Insights:** The previously discovered architectures indicate that leveraging multiple interactions or multiple attempts to identify the most suitable tool can enhance performance (as in Toolformer). Additionally, using a vector similarity approach to retrieve the most relevant tools (as in Toolbench) seems promising.

**Overall Idea:** I propose combining the vector similarity approach with multiple attempts to maximize the chances of selecting the optimal tool. Specifically, I will augment the Toolbench approach by making multiple calls to the LLM to generate several potential solutions and then selecting the best one through a voting mechanism.

**Implementation:** The implementation will involve converting instructions and API documentation into vector representations, retrieving the most relevant APIs, generating multiple responses using the LLM, and finally selecting the best response using a voting mechanism.

```python
class TOOLUSE_TOOLBENCHFORMER():
    def __init__(self, llms_type):
        self.llm_type = llms_type[0]
        self.scenario_memory = {}
        for name, tools in tooluse_IO_pool.items():
            db_path = os.path.join('./db', f'api_pool{name}/')
            self.embedding = OpenAIEmbeddings()
            self.scenario_memory[name] = Chroma(
                embedding_function=self.embedding,
                persist_directory=db_path)
            api_pattern = re.compile(r"\[(\d+)\] ([^:]+):
                (.+?)(?=\[\d+\]|\Z)", re.DOTALL)
            api_matches = api_pattern.findall(tools)
            documents = []
            for match in api_matches:
                api_id, api_name, api_description = match
                first_sentence = api_description.split('.')[0].strip() + '.'
                full_description = f"[{api_id}] {api_name}:
                    {api_description.strip()}"
                doc = Document(
                    page_content=first_sentence,
                    metadata={
                        "name": api_name.strip(),
                        "description": full_description})
                documents.append(doc)
            self.scenario_memory[name].add_documents(documents)
    def __call__(self, task_description, tool_instruction,
         feedback_of_previous_tools):
        similarity_results = self.scenario_memory[task_description].
         similarity_search_with_score(tool_instruction, k=4)
        tool_pool = []
        for idx in range(0, len(similarity_results)):
            tool_pool.append(similarity_results[idx][0].metadata['description'])
        prompt = f'''
You have access to the following tools:
{tool_pool}
You need to select the appropriate tool from the list of available tools
    according to the task description to complete the task:
{tool_instruction}
You must use the tools by outputing the tool name followed by its arguments,
    delimited by commas.
You can optionally express your thoughts using natural language before your
    action. For example, 'Thought: I want to use tool_name to do something.
    Action: <your action to call tool_name> End Action'.
You can only invoke one tool at a time.
You must begin your tool invocation with 'Action:' and end it with 'End
    Action'.
Your tool invocation format must follow the invocation format in the tool
    description.
{feedback_of_previous_tools}
'''
        strings = llm_response(prompt=prompt, model=self.llm_type,
            temperature=0.1, n=3)
        string = self.get_votes(tool_pool, tool_instruction,
            feedback_of_previous_tools, strings)
        return string
```

Figure A.18: New module discovered through AgentSquare search on M3tool.

---

**TH**

**Insights:** From the currently explored architectures, the 'Toolformer' approach seems to have the highest performance at 0.56, which suggests that generating multiple candidate responses and then voting on the best one is effective. Another observation is that a hierarchical search approach like in 'Anytool' might help in better categorizing and selecting tools based on the task.

**Overall Idea:** I'll combine the hierarchical search strategy with the candidate response generation and voting method. This will involve first categorizing the tool based on the task description and then generating multiple candidate responses to select the best one. This should leverage the strengths of both methods.

**Implementation:** I will implement a class where the tool is first selected using a hierarchical search strategy, and then multiple responses are generated for the selected tool, followed by a voting mechanism to identify the best response.

```python
class TOOLUSE_TH():
    def __init__(self, llms_type):
        self.llm_type = llms_type[0]
        self.tool_description = functions_info
        self.tool_pool = travelplanner_toolpool()
        category_prompt = category_prompt()
        string = llm_response(prompt=category_prompt, model=self.llm_type,
            temperature=0.1)
        dict_strings = re.findall(r"\{[^{}]*\}", string)
        self.dicts = [ast.literal_eval(ds) for ds in dict_strings]

    def __call__(self, task_description, tool_instruction,
         feedback_of_previous_tools):
        prompt = f'''{self.dicts}
        You need to select the appropriate tool category from the list of
            available tools according to the task description to complete the
            task: {task_description}
        {tool_instruction}
        You can only invoke one category at a time.
        {feedback_of_previous_tools}
        Output category name directly.
        Your output should be of the following format:
        Category name:
        '''
        category_name = llm_response(prompt=prompt, model=self.llm_type,
            temperature=0.1).split(':')[-1].strip()
        matching_dict = None
        for d in self.dicts:
            if d.get('category name') == category_name:
                matching_dict = d
                break
        if matching_dict and 'tool list' in matching_dict and
             matching_dict['tool list']:
            matched_tools = {tool: self.tool_description[tool] for tool in
                matching_dict['tool list'] if tool in self.tool_description}
        else:
            matched_tools = random.choice(list(self.tool_description.keys()))
        prompt = f'''
        {matched_tools}
        The user's query is :{task_description}
        The tool-use instruction for current task is :{tool_instruction}
        You can only invoke one tool at a time.
        {feedback_of_previous_tools}
        You answer should follow the format: tool_type[tool_arg], such as
            FlightSearch[New York, London, 2022-10-01]
        '''
        strings = llm_response(prompt=prompt, model=self.llm_type,
            temperature=0.1, n=3)
        string = get_votes(matched_tools, tool_instruction,
            feedback_of_previous_tools, strings)
        return string
```

Figure A.19: New module discovered through AgentSquare search on Travelplanner.

---

**CASRC**

**Insights:** The current approaches have explored direct reasoning, step-by-step (Chain-of-Thought), and self-refinement techniques.Notably, the 'Chain-of-Thought' and 'Self-Refine' methods have shown potential by decomposing the task and iteratively improving the solution. However, despite these efforts, the performance still hovers around 50-55%, indicating room for improvement.

**Overall Idea:** To further enhance the performance, I propose combining elements from the high-performing methods (Chain-of-Thought and Self-Refine) with a new focus on.

```python
class REASONING_CONTEXT_AWARE_SELF_REFINE_COT():
    def __init__(self, profile_type_prompt, memory, tooluse, llms_type):
        self.feedback = ''
        self.profile_type_prompt = profile_type_prompt
        self.memory = memory
        self.llm_type = llms_type[0]
        self.tooluse = tooluse
        self.context = ''

    def __call__(self, task_description: str, tool_instruction: str='',
         feedback: str=''):
        if self.memory is not None:
            memory = self.memory(task_description)
        else:
            memory = ''
        if self.tooluse is not None:
            tooluse = self.tooluse(task_description, tool_instruction)
        else:
            tooluse = ''

        # Set up the initial prompt
        prompt = f'''Solve the task step by step. Interact with a household to
            solve a task. Your instructions should follow the examples.
{memory}
{tooluse}
Here is one example.Task: "id": -1, "task": "pddl", "goal": "The goal is to
    satisfy the following conditions: shot1 contains cocktail6. shot2
    contains ingredient1. shot3 contains ingredient2.", "subgoals": ["shot1
    contains cocktail6.", "shot2 contains ingredient1.", "shot3 contains
    ingredient2."], "difficulty": "hard", "additional_info": "subtask":
    "barman"
In this task, the goal is to have 3 shots containing specific ingredients or
    cocktails. This task falls under the "barman" category, which involves
    mixing and serving drinks.

Assuming a bartender robot with a gripper hand, let's break down the steps to
    achieve this:

1. The first subgoal is to have shot1 contain cocktail6. The robot will need
    to locate cocktail6, grasp it using its gripper, and pour it into shot1.

2. The second subgoal is to have shot2 contain ingredient1. The robot will do
    the same as in step 1, but this time locating ingredient1 and pouring it
    into shot2.

3. The third subgoal is to have shot3 contain ingredient2. The robot will
    again repeat the process, locating ingredient2 and pouring it into shot3.

After these steps, the robot will have achieved all the subgoals, thus
    completing the main task.

ATTENTION:You should answer a valid action directly!
Now, here is the task you need to solve:
{task_description}
'''
        # print('prompt:', prompt)
        # input()

        # prompt = prompt.format(task_description=task_description,
            memory=memory, tooluse=tooluse)
        reasoning_result = llm_response(prompt=prompt, model=self.llm_type,
            temperature=0.1, stop_strs=['\n']).replace('>', '').strip()
        reasoning_result = self.refine(task_description, reasoning_result)
        return reasoning_result
```

Figure A.20: New module discovered through AgentSquare search on Pddl.

IR

**Insights:** To maximize the performance of the agent on ALFworld tasks, we should consider incorporating feedback loops and iterative refinement in the planning process. From the discovered architectures, it seems that the most effective modules (DEPS and openagi) provide detailed sub-goals and make use of iterative improvements based on feedback.
**Overall Idea:** Our next planning module will focus on iterative planning with feedback incorporation. After generating an initial set of sub-tasks, the module will prompt the LLM to refine the plan by explicitly checking dependencies and completeness of the sub-tasks.
**Implementation:** We will create a planning module that generates an initial set of sub-tasks and then refines it based on feedback. This refinement will ensure that the sub-tasks are coherent, minimal, and complete, ensuring better performance in sequential decision-making tasks.

```python
class PLANNING_ITERATIVE_REFINEMENT():
    def __init__(self, llms_type):
        self.plan = []
        self.llm_type = llms_type[0]

    def __call__(self, task_type, task_description, feedback):
        few_shot = '''Goal: The goal is to satisfy the following conditions:
            b1 is on b2., b2 is on b3.\nObservation: B1 is on the table. B2
            is on the table. B3 is on the table. Robot arm is empty. The b1
            is clear. The b2 is clear. The b3 is clear.
sub-task 1: {{'description': 'I need to stack b2 on b3 first', 'reasoning
    instruction': 'b2 is on b3', 'tool use instruction': None}}
sub-task 2: {{'description': 'Then I need to stack b1 on b2', 'reasoning
    instruction': 'b1 is on b2', 'tool use instruction': None}}'''

        prompt ='''You are a planner who divides a {task_type} task into
            several subtasks.
First, generate an initial set of subtasks to achieve the final goal.
After generating, refine the subtasks by ensuring they cover all necessary
    steps, are in the correct order, and have no redundancies.
Your output format should follow the example below.
The following are some examples:
Task: {example}'''
        if feedback == '':
            prompt = prompt + '''Task: {task_description}'''
            prompt = prompt.format(example=few_shot,
                task_description=task_description, task_type=task_type)
        else:
            prompt = prompt + '''
end
--------------------
Reflexion:{feedback}
Task:{task_description}
'''
            prompt = prompt.format(example=few_shot,
                task_description=task_description, task_type=task_type,
                feedback=feedback)

        # Initial response
        initial_response = llm_response(prompt=prompt, model=self.llm_type,
            temperature=0.1)
        initial_dict_strings = re.findall(r"\{[^{}]*\}", initial_response)
        initial_dicts = [ast.literal_eval(ds) for ds in initial_dict_strings]

        # Refinement phase
        refinement_prompt = '''
You are an expert planner tasked with refining the following subtasks.
Ensure all necessary steps are covered, they are in the correct order, and
    there are no redundancies.Your output format should follow the example
    below.
The following are some examples:
Task: {example}
end
--------------------
Subtasks: {subtasks}
'''.format(subtasks=initial_dicts,example=few_shot)

        refined_response = llm_response(prompt=refinement_prompt,
            model=self.llm_type, temperature=0.1)
        refined_dict_strings = re.findall(r"\{[^{}]*\}", refined_response)
        refined_dicts = [ast.literal_eval(ds) for ds in refined_dict_strings]

        self.plan = refined_dicts
        return self.plan
```

Figure A.21: New module discovered through AgentSquare search on Pddl.

