# OpenReview forum: "AgentSquare: Automatic LLM Agent Search in Modular Design Space"
_ICLR.cc/2025/Conference — ICLR 2025 Poster_

### Official Review · Reviewer_Dm8t · 2024-10-26

**Soundness:** 3
**Presentation:** 2
**Contribution:** 3
**Rating:** 6
**Confidence:** 5

**Summary:**

I am disheartened to report I believe this paper involves a form of plagiarism and likely involves intentional academic misconduct (or at least suspect behavior). For full transparency, and because I am an author of the plagiarized work and thus have a stake beyond a normal reviewer, I am making my review public and non-anonymous.

We posted a paper called “Automatic Design of Agentic Systems” (ADAS) on August 15th to arXiv and shared it on X and LinkedIn. It was submitted for review at ICLR 2025.

This review is of AgentSquare: Automatic LLM agent search in modular design space. It was posted to arXiv on Oct 8 2024.

This AgentSquare paper clearly takes many components from the ADAS paper, including the main ideas (slightly modified), repackages them (after adding a bit of new work), and presents the paper as a completely new set of ideas, all without acknowledging the significant degree to which it has lifted many key elements from our paper. It does cite ADAS, but only as an afterthought in the last paragraph before the conclusion, rather than writing throughout the paper (including in the motivation) that the ADAS paper is an extremely relevant piece of prior work and from which they copied (and slightly adapted) core pieces: the main ideas, unique writing choices, data visualizations, code, and prompts (which have significant chunks directly copied without modification).

It is hard to believe this is an honest mistake. Instead, the evidence causes me to conclude it is an attempt to get an ICLR publication by hoping the reviewers are not aware of how similar this paper is to a previous arXiv paper. Given how similar this new piece of work is, I would expect (and feel scholarship demands) this new paper to mention ADAS early and often, including discussing how it inspired their work and how the works differ, and to acknowledge that this new paper uses many pieces of the ADAS paper. I would also expect AgentSquare to compare to ADAS, since a central claim is that it is an improvement over the approach ADAS took. Instead, the paper is written in such a way that a busy reviewer/reader might not know such a similar work exists, and thus mistakenly attribute credit for the big ideas, results, and novelty to this new work.

If the authors are somehow so unaware of what is appropriate and required in academic writing that this is all an honest mistake, then in my view they should let the community know that, apologize, and entirely rewrite the paper to properly inform readers.

Note: this is almost certainly not a case of the work being done concurrently, because the paper copied so many pieces from ADAS (including prompts used in their experiments).

To be clear, there are some nice new innovations in the paper (if the data are to be believed, see below). Properly written (and with more careful comparisons to ADAS and ablations of what makes this paper different and if they help), this paper could have been accepted as a nice innovation on top of ADAS. Instead, I believe it should be rejected as plagiarism. There is not enough time and rounds of review in this conference's review process for reviewers to consider a full rewrite (including back and forth) at this stage.

Here is evidence of untoward behavior:

1. The prompts from ADAS are largely copied verbatim and then modified in a few places. The amount of overlap makes it certain they copied from our work, yet they did not acknowledge doing so. They should report using a prompt from another paper. But more importantly, it shows how similar the work is to, and was inspired by, ADAS, and thus that ADAS should have been mentioned throughout (as prior work, something to be compared to, something they built on and were inspired by, etc.). You can see the similarities here: https://drive.google.com/file/d/1vHPW2EXvx7LjFv-kDhQHyb_VGHTnndD8/view?usp=sharing

2. ADAS did something rare: it said it was (A) recognizing and naming “a new research area” named Automatic Design of Agentic Systems, and (B) introducing a new search algorithm within that area named Meta Agent Search. AgentSquare clones that template, saying “we introduce a new research problem: Modularized LLM Agent Search (MoLAS)” and “building on this design space, we present a novel LLM agent search framework called AgentSquare.” It is uncommon for papers to name a new research area explicitly, let alone also introduce an instance within it.

3. Moreover, MoLAS is not a new research area. It is a subset of ADAS. It is a more constrained subset of ADAS, at least the way it is initially presented, which is that MoLAS is constrained to combine pre-existing, human-designed agent modules. However, in a contradiction that is confusing and harms novelty, later the paper also says the modules can be changed, removing this constraint, meaning that MoLAS is thus just ADAS with a new name?

4. Even if ADAS is not identical, it clearly is a very related prior work, yet it is not mentioned in the introduction, nor anywhere in the paper until the second-to-last paragraph. Yet other, much less related work is mentioned (line 053), suggesting an intentional choice to deceive the reader/reviewer by hiding work that hurts the novelty claims of this paper. Also, some of the claims in the introduction say effectively that things like ADAS do not exist or are rare (line 047): clearly ADAS should be mentioned in that context, but isn’t.

5. The plot style of ADAS Fig. 3a is very similar to Fig. 4, providing further evidence of the central role the ADAS paper played in inspiring and catalyzing the AgentSquare paper.

6. One (somewhat, see * below) new thing in AgentSquare vs. ADAS is an explicit surrogate model that predicts an agent’s performance, saving compute vs. running a real evaluation. However, I am very skeptical of this data. Even with much training and years of research, getting surrogate models to be highly predictive (e.g. in Neural Architecture Search) is very difficult. I have a hard time believing an LLM is even good zero shot, yet these data say it is nearly PERFECT zero shot. That raises an important question: why are there so few (5) points shown per run (e.g. versus the many more shown in many more per run in Fig. 4)? How did you select these 5? Can you provide code that replicates this experiment, so the community can run it independently and verify it? I do not mean the overall search, which is expensive, but just the zero-shot predictions and the evaluations for all agents produced per experiment. I have never asked for such a thing in review before, but extraordinary claims require extraordinary verification, in my mind, especially given the cloud of suspicion raised by all the other issues in this review.  If the surrogate model is indeed this good, that is a major discovery and worthy of publication in some form, and I recommend you share it in a properly written paper.

The net effect of all of this evidence forced me to conclude the paper is at best deeply flawed due to many innocent mistakes, and at worst intentionally dishonest, with the latter seeming far more likely.

If there were no plagiarism issues, I would also point out:

- There is no ablation of memory, one of the key new things they claim they added, and thus no evidence it helps

-  The paper should cite https://arxiv.org/abs/2206.08896

- *Surrogate models arguably implicitly exist within ADAS since it asks for proposals the model thinks will be high-performing, meaning it could use its predictions to guide proposals. However, we know from things like Chain of Thought and Reflexion that one can coax better performance by asking an LLM in the right way and/or asking it to reflect post-hoc on a creation.

- The paper claims that ADAS’ search in the space of code is an over-simplification. While it is true that describing the search space is simple, searching in it is MUCH more complex. I think it is much more appropriate to say that MoLAS is simplifying search by constraining it to a simple, small search space vs. ADAS, the latter of which allows any possible agent.

- The paper claims that ADAS is proposed without consideration of existing agent architectures. I disagree, since one can easily inject them as seeds in the search archive (and we did consider that). But we prefer that to be a choice: it likely speeds up search, but also biases it.

It saddens me to make these accusations. However, I deeply believe in the integrity of the scientific process, including peer review. It relies on trust, and I think we need to stay vigilant and hold people accountable so we can maintain that trust as much as possible.

Jeff Clune

Professor, Computer Science
University of British Columbia
JeffClune.com

Canada CIFAR AI Chair & Faculty Member
Vector Institute

**Strengths:**

See main review.

**Weaknesses:**

See main review.

**Questions:**

See main review.

**Details Of Ethics Concerns:**

See main review. Also note that because I have am a co-author on the paper I believe is plagiarized, and thus have a conflict-of-interest, I have added my name to my review. I hope that is alright, as it seems more ethical to me than otherwise. Please let me know if you have concerns and we can discuss the best path forward.

[Edit: Please read the entire exchange. By the end, the authors have fixed the issues I raised.]

---

> ### Comment · Reviewer_Dm8t · 2024-11-14
> **Do the other reviewers agree this  a form of plagiarism and likely involves intentional academic misconduct (or at least suspect behavior)?**
>
> Do the other reviewers agree with my assessment that this paper is a form of plagiarism and likely involves intentional academic misconduct (or at least suspect behavior)?
>
> I am sorry to ask since I know everyone is busy, but since you have already read the paper I am hoping reading my review and offering your reaction is not too large a time burden. Given the stakes I think it is worthwhile (but of course I understand we are all juggling many demands).

---

> ### Author Response · Authors · 2024-11-15
> **Direct evidence against the false accusation of idea plagiarism**
>
> The research idea of our project was proposed on April 27th. It is entirely original, and well before the release of ADAS (paper on August 15th, code on August 19th) Our key innovation is to propose a modular design space and a search algorithm within it. It allows researchers easily combine classic agent designs like assembling computer parts and discover new modules can be directly reused by others. It took us a long time to finish the project because the huge workload of abstracting all the classic agent designs into a standardized modular design space. ADAS is a code/prompt optimization framework that has no such modular feature.
>
>
>
> We provide the following **Direct Evidence** to support our statement:
>
> 1. **A video of a public presentation of our work on August 22nd.** It is broadcasted, recorded and timestamped by commercial remote meeting service, and stored in their cloud server since that day.
> 2. **A seminar notification email sent on August 20th.** It includes an abstract that clearly states the modular search idea and experiment results.
> 3. **Extensive raw communication logs extracted from commercial Instant Message application.** There are organized as a timeline, showing our communication groups is created on April 27th, discussion about modular design space on the first day, first mention of modular search on May 7th.
>
>
>
> These materials can be accessed at https://www.dropbox.com/scl/fo/bb3tjrbbicv0mo6u7ic8h/ALBWPoIQjiaMuxXwD0aNuoM?rlkey=p26l2l7qmx1tz07zfrehhw36s&st=0gp6z0uu&dl=0. They are direct and decisive evidence that refute the false accusation of idea plagiarism. Our key idea is to propose a modular design space of LLM agents, which is inspired by the agent formula of Lilian Weng as described in Line 138-139 in our paper, rather than those code/prompt optimization works like ADAS, Promptbreeder, OPRO, etc. This evidence shows that our research idea was conceived and developed independently from ADAS, and any claims that AgentSquare was "copied verbatim" are entirely false.
>
>
>
> In addition to above evidence, we encourage ACs and other reviewers to further read the papers of ADAS and AgentSquare. We know it is a big ask for your valuable time, but we believe you can reach fair assessment after reading both papers in first hand.
>
>
>
> Note: To keep the integrity of blind review at this stage, we provide anonymized screenshots of recorded video and communication logs. The link to original video in cloud server has been sent to ACs, which can be available with their permission and made public after the review process. We can also provide the raw communication logs proving how we form and implement our idea of modular agent design starting from April 2024, should further evidence be required.

---

> ### Author Response · Authors · 2024-11-15
> **Response to the so-called “evidence” listed by Reviewer Dm8t**
>
> ***Open-source code reuse:*** We must point out the exaggeration and misrepresentation in Reviewer Dm8t’s claim *"The prompts from ADAS are largely copied verbatim and then modified in a few places."* The fact is that we only refer to a very small fraction of prompts from ADAS's open-source codes of querying LLMs for code writing. Completely replacing these prompts would not impact our method’s functionality entirely. Unlike the one-sided prompt comparisons you show, we conduct a comprehensive comparison of codes between the two works via JPlag (https://github.com/jplag/JPlag), an advanced source code overlap detector. The result is presented in https://www.dropbox.com/scl/fo/xjv4t4ltdininc79ifdmt/AEGhqQe06iT05tgbu24TWqc?rlkey=txgx5tfs5e0843okcnply6evd&st=qt89u6x1&dl=0, revealing that **the overall code similarity with ADAS is only 1.8%**. Reusing parts of open-source codes and prompts is a standard practice in the community and should not be used as evidence for idea plagiarism. Additionally, we also find part of prompts in ADAS has overlap and is modified from previous projects like DiscoPop (https://arxiv.org/abs/2406.08414, released on June 12), and the overall idea is very similar to existing code/prompt optimization works such as DiscoPop and OPRO, all following a "meta-prompt -> LLM -> output code/prompt -> evaluation" pattern. Below are some examples of prompt reuse of ADAS from DiscoPOP:
>
>
>
> Example 1 (the output instruction prompts):
>
>
>
> ADAS: *"**The first key should be ("thought"), and it should capture your thought process for designing the next function.** In the "thought" section, first reason about what should be the next interesting agent to try, then describe your reasoning and the overall concept behind the agent design, and finally detail the implementation steps. **The second key ("name") corresponds to the name of your next agent architecture**. **Finally, the last key ("code") corresponds to the exact** **“forward()” function in Python code that you would like to try.**"*
>
> DiscoPOP: *"When you respond, output a JSON where **the first key ("thought") corresponds to your thought process when designing the next function**. **The second key ("name") corresponds to the name of your next function.** **Finally, the last key ("code") corresponds to the exact python code that you would like to try.**"*
>
>
>
> Example 2 (the same output format prompts: "thought", "name", "code"):
>
>
>
> ADAS: {***"thought":** "Insights:\nYour insights on what should be the next interesting agent.\nOverall Idea:\nyour reasoning and the overall concept behind the agent design.\nImplementation:\ndescribe the implementation step by step.",*
>
>   ***"name":** "Name of your proposed agent",*
>
>   ***"code":** """def forward(self, taskInfo):*
>
>   *# Your code here*
>
>   *return answer*
>
> *"""*
>
> *}*
>
> DiscoPOP: *{**"thought":** "Based on the previous outputs, I should try the direct preference optimization algorithm.",*
>
> ***"name":** "dpo",*
>
> ***"code":** "def sigmoid_loss(...)"*
>
> *}*
>
>
>
> These all suggest that ADAS is built on prior frameworks, however, the ADAS paper does not mention their idea comes from DiscoPop or refer to DiscoPop when introducing their design. Does this imply reusing open-source code/prompt is a common practice of the community?
>
>
>
> ***About the doubt of proposing a new research area:*** Our key innovation is to propose a modular design space of LLM agent, focusing on the compositional structure of LLM agents, which sets it apart from the research problems of prompt/code searching like ADAS, OPRO and Promptbreeder. It is only natural to propose a search algorithm in this new design space as technical contribution. It is impossible for Reviewer Dm8t to claim copyright for proposing new research problem.
>
>
>
> ***About "MoLAS is only simplification of ADAS":*** MoLAS searches in a smaller space compared to the entire universe of possible agents defined in python code. But we will argue it is also a more fruitful space since search algorithms can swiftly recombine successful classic design, and allow open-endedness to explore new modules. One analogy will be fully-connected MLPs can theoretically approximate any function, but more structured architectures like CNNs are more successful in CV. Anyway, the difference in search space should be considered as a sign of our original contribution.
>
>
>
> ***About the plot style similarity:*** Plotting search trajectory is very common in any paper about search problem, which can also be found in many works on "LLM for search" [1] [2] and AutoML [3].
>
>  [1] Lu, C., et al. Discovering Preference Optimization Algorithms with and for Large Language Models. *arXiv preprint arXiv:2406.08414*.
>
> [2] Shojaee, P., et al. Llm-sr: Scientific equation discovery via programming with large language models. arXiv preprint arXiv:2404.18400.
>
> [3] Real, E., et al. Automl-zero: Evolving machine learning algorithms from scratch. In ICML 2020.

---

> ### Author Response · Authors · 2024-11-15
> **About breaking the double-blind review protocol**
>
> In the end, we must emphasize that **revealing personal identifiers, including the reviewer’s name and professional title, severely violates of ICLR’s double-blind review policy**. Reviewer Dm8t also chose to include a list of big titles, which is totally unnecessary even for reporting misconduct. It imposes unfair influence and pressure on other reviewers, ACs and public readers. Reviewer Dm8t also has clear conflict-of-interest of reviewing this paper.
>
>
>
> We stand by the originality of our work, the integrity of our research timeline, and our adherence to academic norms. We will fight for our work’s undeniable originality, and expect a fair and objective evaluation based on these facts.
>
>
>
> Sincerely,
>
> Anonymous Authors

---

> ### Comment · Reviewer_Dm8t · 2024-11-19
> **Cementing my view of academically inappropriate behavior, the authors' reply has no recognition of clear problems, no offers to fix them, and no apologies**
>
> The authors were notified that their paper:
>
> 1. Does not properly inform the reader of an important prior work (and one that the authors themselves built off of), which should be fixed in order to make the paper non-deceptive to the reader, let alone publishable. ADAS introduces the idea of searching for LLM agents with LLMs. Despite this prior work it built upon, AgentSquare claims to introduce the idea of searching for LLM agents with LLMs, albeit in a modular way, and does not inform the reader (especially in the first 90% of the paper) that ADAS already does that.
>
> 2. Copies and uses large parts of core methods from that work, without attribution, which should be fixed.
>
> 3. That the above is academically inappropriate at best, and plagiarism at worst.
>
> 4. Claims to improve over ADAS, though oddly without mentioning ADAS (i.e. claims that a modular search space is better than a less-constrained code space), yet does not offer any experiments to validate these bold claims, which are central to the paper. It needs to perform such experimental comparisons to ADAS in order to be publishable.
>
> The authors did not respond with a single offer to fix these issues, nor any apologies. I believe all of this is disqualifying for publication. Moreover, I believe it should be flagged by reviewers and ACs as academically inappropriate. Even if one does not think there are issues of plagiarism or academic dishonesty here, at a minimum there are many changes my review should have prompted, yet the reviewers apparently are unwilling to fix any of these issues.
>
> I will now reply to the issues the authors replied to. However, I point out that other issues remain and were not addressed, so anyone reading this should make sure to read my original review before deciding for themselves.
>
> On the Prompt Copying:
>
> 1. You agree that you copied and used our prompt, with large chunks of it (many paragraphs) mostly or completely unmodified
>
> 2. Do you not think that should be mentioned in your paper?
>
> 3. Moreover, does that not add evidence to the fact that your paper was heavily inspired by ours?
>
> 4. If so, do you think the paper informs the reader of that fact?
>
> 5. If not, do you think it is ok that it does not?
>
> On some of your other points:
>
> 1. You write: “The fact is that we only refer to a very small fraction of prompts from ADAS's open-source codes of querying LLMs for code writing. Completely replacing these prompts would not impact our method’s functionality entirely."
>
> It does not matter (in terms of properly citing work in academia) whether your algorithm might work just as well without the very large chunk of text that was copied. Instead, the issue is that you did copy and use that text, but did not say so. Moreover, this is additional evidence in the larger story that your paper was heavily inspired by our paper, yet that is not at all communicated to the reader. See my original review on that point.
>
> It also does not matter that the large section of copied text is a small fraction of your code base. I can write a 1000 page novel, but if I copy two pages from someone else without attribution, that’s still plagiarism.
>
> In addition, the claims that “only a very small fraction of prompts are referred to from ADAS” and “the overall code similarity with ADAS is only 1.8%” are misleading. The copied prompt serves as the central and foundational prompt for evolving all four agent modules in the AgentSquare paper. Moreover, given that ADAS and AgentSquare include extensive code for different experimental domains, comparing the overall similarity of the entire codebases is not informative.
>
> More informative than the entire code-base comparison is to note that there are four* (!) separate prompt files (all central to the ideas of AgentSquare), that, using the same plagiarism tool the AgentSquare authors used, have similarities to the ADAS file “mmlu_prompt.py” of 76.6%, 65.9%, 68.9%, and 65.9% respectively. There are also identical or similar code structures and variable names in these files. Additionally, one can visually compare ADAS prompts to the generic prompt from the AgentSquare appendix to see substantial overlap in this image: https://drive.google.com/file/d/1vHPW2EXvx7LjFv-kDhQHyb_VGHTnndD8/view
>
> *prompt_reasoning.py, prompt_memory.py, prompt_planning.py, prompt_tooluse.py
>
> 2. You write “Reusing parts of open-source codes and prompts is a standard practice in the community and should not be used as evidence for idea plagiarism.”
>
> The issue is not using others’ open-source code, but (1) doing it for the core ideas in your paper, and (2) passing off that work as your own original, meaningful contribution. The correct scientific convention is to report when you build on others’ contributions, especially when that is core to the idea (here, LLMs automatically designing LLM agents).
>
> [post 1 of 3 of this reply]

---

> > ### Comment · Reviewer_Dm8t · 2024-11-19
> > **Part 2/3 of my reply "Cementing my view of academically inappropriate behavior..."**
> >
> > [part 2 of 3 of my reply]
> >
> > 3. Re: ADAS reused prompts from DiscoPOP. This is a false equivalence. First, the 1-2 sentences are simply for formatting the LLM's outputs, which is not a core idea in our paper. In the case of AgentSquare, more than a page of prompts are taken that serve as the central prompt for the algorithm and represent the core of the idea in the paper (LLMs automatically designing LLM agents). Second, the ADAS paper cites DiscoPOP and properly describes its relation to ADAS, which is not true with AgentSquare vs. ADAS. Third, the CoT-style “thought-answer” output formatting pattern mentioned above is common in agent-related work. It was thus an honest mistake that we did not mention our use of those prompts. Now that we are aware, we recognize we should make that explicit, and will update the paper immediately. Our attitude is thus in stark contrast to yours in terms of acknowledging and rectifying the issue.
> >
> > 4. You write: “It is impossible for Reviewer Dm8t to claim copyright for proposing new research problem”. Of course, I am not claiming that. The problem is that you knew of our paper, and your paper adopts this exact template (of both ideas and writing), yet barely mentions our paper *despite the extreme similarity*, and the mention is only to dismiss it as barely relevant in the penultimate paragraph. It is one piece of evidence that paints a larger, troubling picture, as described in my original review.
> >
> > 5. MoLAS is a subset of the ADAS problem, and needs to be articulated as such. You explicitly acknowledge this fact in this review process, but fail to do so in the paper. You quoted the following from my original review: “MoLAS searches in a smaller space compared to the entire universe of possible agents defined in Python code.” To that, you replied: “But we will argue it is also a more fruitful space since search algorithms can swiftly recombine successful classic design, and allow open-endedness to explore new modules. One analogy will be fully-connected MLPs can theoretically approximate any function, but more structured architectures like CNNs are more successful in CV. Anyway, the difference in search space should be considered as a sign of our original contribution.”
> >
> > Given all of that, MoLAS should have been motivated by ADAS. In other words, you should inform the reader that ADAS exists, but is unconstrained, and then say you have a new algorithm MoLAS that is an improvement over ADAS because MoLAS is constrained and therefore better. Additionally, given what you wrote, MoLAS should be experimentally compared to ADAS since you claim it is better (a key claim of your paper). Your analogy is great: it IS like CNNs vs. MLPS, **which is why CNNs were compared to MLPs to show they were better!!** I feel like you are making the case against how your paper dealt with things in your reply here. As I wrote in my review, what you did would count as an original contribution had it been properly motivated and with proper credit being given. But instead, your writing deceives the reader into not knowing that extremely relevant prior work exists (and that you built on it!). Irrespective of any claims of plagiarism, as a reviewer I believe such a comparison should be performed and would guess most other reviewers would feel the same once they know about ADAS and its similarity (which is something your paper greatly obfuscates for the reader).
> >
> > 6. Your timeline. The issue is not when you had the ideas and worked on them. The issue is how you wrote the paper given that you were aware of the ADAS paper, especially since you built off of the ADAS paper to conduct your experiments. It deceives the reader by not informing them about such similar prior work that you built on. The reader is trusting you to know this area and fairly report on it. It seems like you are defending that choice and do not believe it necessary to modify your paper. Is that right? If so, I do think that is academically dishonest.
> >
> > I also want to clarify some things:
> >
> > 1. I included my name for full transparency because the paper plagiarized work by my co-authors and I, which inherently creates a different status for me versus an uninvolved reviewer. I believe including my name is thus the ethical thing to do. I informed the ICLR program chairs that I was doing that and asked if that is appropriate (given their policy). They asked me to leave my name in and that they would contact me if after further discussion they thought I should remove it. They never did, so it remained in per their instructions. The default policy is not meant for this case, and they thus seemingly concluded that an exception to that policy was the appropriate action in this case.
> >
> > 2. I also included my titles for transparency. In any case, they are easily discovered via an online search. I also believe they are inconsequential and that people will judge the case on its merits.

---

> > > ### Comment · Reviewer_Dm8t · 2024-11-19
> > > **Part 3/3 of my reply "Cementing my view of academically inappropriate behavior..."**
> > >
> > > [part 3 of 3 of my reply]
> > >
> > > Overall, I am saddened by the tone you chose to strike in your response. I believe the academically appropriate thing to do in this case is to apologize, withdraw the paper, and rewrite it to fix the issues raised in my review before submitting it elsewhere for publication.
> > >
> > > I also think the AC, reviewers, and ICLR leadership should read all the evidence and form their own opinions, as I think after doing so they will agree.

---

> ### Author Response · Authors · 2024-11-21
> **Further Response to Reviewer Dm8t (1/3)**
>
> In the new comments, the reviewer complains about the "academic inappropriateness" of credit attribution, and we "did not respond with a single offer to fix these issues." We would like to clarify that we were facing serious idea plagiarism accusation in first round of review, so our natural priority would be to present all the facts along the timeline of our research projects (https://www.dropbox.com/scl/fo/heev3q81p28cj4wu6dpnp/AJ6F5TZRVaq0J7h4EjA-ze8?rlkey=nhwgs21y9h7n1a049t7ign7b7&st=uf0fx0qf&dl=0), proving our work has a clear independent origin before we are aware of ADAS.
>
> In the meantime, we have taken steps to fix the reviewer's other concerns. We now are able to provide further response to address them:
>
>
>
> **About the inspiration of our work:**
>
> You complain our paper does not "communicate to the reader" that our work is "heavily inspired" by ADAS. The word "inspired" implies the idea of our work comes from ADAS, which will be untrue for us to say so. As we show in the list of evidence (https://www.dropbox.com/scl/fo/heev3q81p28cj4wu6dpnp/AJ6F5TZRVaq0J7h4EjA-ze8?rlkey=nhwgs21y9h7n1a049t7ign7b7&st=uf0fx0qf&dl=0), our communication logs for this project started from April 27th, we independently proposed the idea of modular search on May 7th,fy or "built off of" your ADAS project with its code released on August 19th.
>
> Throughout our project, we were inspired by Lilian Weng's concept of modular agent design. We propose an operational implementation of the modular agent design space, facilitating the search algorithm on this space. We accurately described this in our original paper (*"Many experts in the field have proposed building LLM agentic systems with key modular components from engineering (Weng, 2023) and cognitive perspectives (Sumers et al., 2023; Shao et al., 2024). However, these proposals remain largely conceptual, lacking implementable solutions to unify existing LLM agents."* in Section 2).  It will be unfair to them if we misrepresent the inspiration of our work.
>
> This timeline also clearly shows that our work is not started after we are aware of ADAS. Therefore, we did not mislead readers/reviewers regarding the origins of our idea and the motivation of our work.
>
> Since you were probably unaware of our research timeline and all the supporting evidence in initial review, we can partly understand you raise suspicion of us "repackaging" your main idea. We think you were also uncertain at that time. Otherwise, you won't list several other points in case "if there were no plagiarism issues" or asked for other reviewers' opinions. **Now that all the evidence of our research timeline have been clearly presented, we hope you would respect these facts and acknowledge the independent origin of our research idea.**

---

> ### Author Response · Authors · 2024-11-21
> **Further Response to Reviewer Dm8t (2/3)**
>
> **About the used prompt:**
>
> You argue "ADAS reused prompts from DiscoPOP" is a "false equivalence" to our work reused prompts from ADAS, because we "do so for the core idea" of our paper and ADAS didn't. We disagree the prompts we used is the core idea of our work. We respond to your complaint from the following four points:
>
> (1) **About the code comparison.** You reported our 4 files have 65%+ overlap with  “mmlu_prompt.py”  in ADAS based on the detection software. We found you may have misused this software as we tried to reproduce this result (see the overlap detection we reproduce: https://www.dropbox.com/scl/fo/9uunc4x4vyj5y9vj3ekwo/ANBkcG2x0a8T720p6kE8LxI?rlkey=hl12odwalyuye2bmc0tznrapi&st=d73g0s49&dl=0). This result identifies everything in the *{'thought': , 'name': , 'code': }* dictionary as overlap, only because they have same key names. It is a wrong comparison, because the dictionary structure and key names are originally designed in DiscoPOP for output control. It should not be used to identify the overlap between ADAS and AgentSquare. More importantly, the majority of the content in these dictionaries are the standardized modules we wrote independently, which should not be identified as overlap. Using a more accurate word-by-word analysis mode, the software reports an overlap rate of 8.01%-14.73% (see the detection report: https://www.dropbox.com/scl/fo/4ep4peni3uvoyelcjmfi6/AE-chq5_WC5tsY6J7iA4W-U?rlkey=gp1izapky19vx6f4h95tllprk&st=q3b4haxp&dl=0). The remaining 85%~92% are used to define modular search space and enabling search algorithm to produce modules with standard IO interface, which are part of the core idea of our work, as we clearly stated in our Intro "*the **core of our work** is a modular design space for LLM agents*". Thus, they should not be disregarded.
>
> (2) **About our core idea**. As we mentioned in last point, we have clearly stated in our Title and Intro multiple times that our core idea is to propose a modular design space and a modular search algorithm on top of it. To support this core idea, the important code files include those under "modules", "module_recombination" and "module_predictor" folders and "workflow.py" in our repo. These files implement the function of modular design space, module recombination and performance prediction. You also did not have dispute that these are our original contribution in initial review.
>
> (3) **About the reason we use your prompt**. We were originally motivated by FunSearch to ask LLM to optimize the designs of each module. We consider asking LLM to improve prompt/code a general procedure reported in many previous works, such as Promptbreeder, OPRO, and FunSearch, some of which are also noted in ADAS paper "Although a few existing works can be considered as ADAS methods, most of them focus only on designing prompts (citing Promptbreeder; citing OPRO)." After building the modular search framework, we did use part of prompts from ADAS as a good implementation of this general procedure in module evolution function. However, please also note the prompt we used will not work independently for module evolution, since they are not constrained and guided to generate modules with standard I/O interface that allow them to cooperate with other modules in our designed workflow. Therefore, we wrote extensive original code/prompt for modular template and workflow to guide LLM to generate law-abiding modules with standard I/O interface (workflow & template and module examples: https://www.dropbox.com/scl/fo/h6j8ppbwvykxoivvgar7d/AI-Rzr8dnZwlIIxkbJHz_Hg?rlkey=qq75xbl48ltu0na56chn03lll&st=7p1l9avk&dl=0), which is a key design for module evolution to work. The reasons we think it is OK to use the part of open-source prompts from ADAS are that: [a] it does not affect our core idea of modular search; [b] we are not using it to claim ADAS's innovation of searching "entire agentic system in code".
>
> (4) **We never intend to deceive anyone the used prompts are our core idea**. We genuinely believe the piece of prompts we used does not affect the core idea of our work. Otherwise, we would not put the unmodified version in open repo. It will not be difficult if we want to rewrite them since the overall percentage is low and they are largely natural language. Besides, throughout the review process, we have been very transparent about reusing this part of prompts from ADAS's open-source repo.
>
>
> **How we propose to fix:**
>
> (1) You argue it will be more academically appropriate to explicitly disclose the use of other's prompt in paper, and you are willing to do so for reusing prompts from DiscoPOP. We appreciate your transparency and rigorous practice about that. We have also clearly stated that we use part of prompts from ADAS in module evolution section.
>
> (2) We can also rewrite the prompt from scratch, if you think that is necessary. We will replace all results of AgentSquare with the new prompts.

---

> ### Author Response · Authors · 2024-11-21
> **Further Response to Reviewer Dm8t (3/3)**
>
> **About the credit attribution:**
>
> Regarding credit attribution to ADAS, we want to first point out that we have cited and discussed it in the related work section: *"Thirdly, there have been some separate works to automate parts of LLM agent systems, such as prompt optimization [some works] and workflow search [your work]. Nevertheless, these works over-simplify the structure of agentic systems to pure prompts or codes, unable to provide a clear design space of LLM agents"* Although you may disagree with our judgement (and it is discussable), we did discuss our work's relationship to your ADAS paper, and present our key contribution as a new design space.
>
> Your concern that it appears too late actually stems from our paper organization with the related work as the second-to-last section. We chose this arrangement because we thought the concepts of modular design space are more relevant to our key contribution, and we put them in the second section.
>
> **How we propose to fix:** (1)  We have strengthened the discussion of ADAS in the second paragraph of Introduction and Related Work. Besides, we add a new paragraph at the beginning of Background. (2) We have run ADAS on our benchmarks and added it as a baseline to further address your concern. Specifically, we keep experiment settings (i.e. task description for LLM, number of iteration) as the same with all other searching methods. We have updated these results in the revised paper.
>
> If you have other reasonable concerns about credit attribution, we will also consider in our further action.
>
>
>
> **About our tone in last response and double-blind review rule:**
>
> We would like to say that we have tried to be as polite as we could. However, in the first round of review, we were wrongfully accused of "idea plagiarism", thus it is hard for us to do so. We feel sorry if you think our tone is not good enough.
>
> As for the disclosure of your name and professional titles, if the PCs instructed you to do so, we will respect their decision.

---

> ### Author Response · Authors · 2024-11-25
> **Link Update**
>
> We noticed there might be some technical difficulty to access the google drive links we previously uploaded. We have replaced them with dropbox links. Please check out our uploaded materials and kindly let us know if you have any trouble accessing them. Thank you!

---

> ### Comment · Reviewer_Dm8t · 2024-11-27
>
> [Response to the response 1/2]
>
> I appreciate that the authors are finally softening their tone, and I feel we are starting to understand each other. Critically, I am also happy to see that the authors are making changes to the paper to address many of the points I have raised. The tone of the replies is a bit surprising because it lacks an apology or recognition that things were not done correctly, but the edits to the PDF are in line with those things, and have moved the PDF substantially toward the direction I think is appropriate. Overall, as I have said from the start, the paper does make a nice contribution and is worthy of publication if it properly describes prior work and what it built off of. I think the current version of the PDF is close to accomplishing that. I am willing to substantially increase my score if the remaining issues I believe exist are addressed.
>
> Here are my thoughts at present:
>
> It does help to know that you had the idea independently of ADAS. But even given that, since ADAS was shared first, and you knew about it, it is not appropriate to then write most of the paper as if ADAS does not exist. In science, once a paper comes out that is relevant, it should be described, so the reader is aware of that relevant work. The original manuscript gave the impression that you are proposing **for the first time** the use of FMs to design Agentic Systems. That is false, and thus deceived the reader, and did not give appropriate credit to ADAS, which shared that idea prior to AgentSquare. You say your paper did mention ADAS. That is true, but (1) the abstract intro and entire paper gave the impression I just described. Many people do not read all the way through to the final paragraph, so it is not acceptable in the abstract and intro (which set the stage motivating the work) to ignore something so related. (2) The sentences about ADAS in the original ms did not convey just how related ADAS is, and incorrectly said “to automate parts of LLM agent systems”, (note the key word “part”) implying that ADAS automates a small part, whereas AgentSquare does something more (which is not true). But even with these sentences, a reader would be rightly confused if in fact there were such a related piece of work, yet it was only quickly described in 1-2 sentences just before concluding…whereas the rest of the paper pretends that work does not exist. They thus likely would come to the wrong conclusion despite those sentences toward the end. Sometimes it is ok to have a section called related work at the end that discusses tangentially related work. But in this case, the current organization misleads the reader/reviewer because there is a piece of related work that is so central. Moreover, your main claim should be that your system is better than ADAS because it is modular (see the next point), and that thus needs to be in the motivation section to set up your experiments, but none of that happened in the original manuscript. All of that said, I do think the updated PDF handles this correctly. Thank you for fixing it.
>
>
> Relatedly, regarding being “wrongfully accused of "idea plagiarism", I stand by that claim as regards the original paper. As I wrote above, even if you had the idea independently, once another paper comes out, it is not ok to write the paper in such a way as to ignore the other paper and make a reader or reviewer believe you are proposing an idea for the first time, when you know that idea has already been shared publicly. That is still idea plagiarism, since you are misleading people into believing you are the first to propose an idea. That is what your original draft did, even though you did mention ADAS toward the very end in a cursory way. However, this issue has been fixed in the revised PDF, which I am glad to see. Thank you for fixing it.
>
> Regarding the 2nd sentence you quote from the original paper “Nevertheless, these works over-simplify the structure of agentic systems to pure prompts or codes, unable to provide a clear design space of LLM agents.” reveals yet another major flaw in the original paper. This sentence IS what SHOULD BE the central claim in the paper: something like “ADAS exists, but we introduce a more modular space, and that is a better (higher-performing) choice.” That motivation for a paper would be fine (as I said in my original review), but then of course requires (1) setting up the motivation that way and (2) doing experiments to validate that central claim. Fortunately, the revised paper does exactly that.
>
> [see next reply for the rest of my reply]
> [Response to the response 1/2]

---

> > ### Comment · Reviewer_Dm8t · 2024-11-27
> >
> > [response to the response 2/2]
> >
> > I disagree you cannot compare to ADAS. Both are searching for agents to solve tasks. You argue it would be unfair to ADAS because AgentSquare would be more sample efficient, but that is exactly your central claim! So it is essential to show it.  I also am confused by the contradiction of saying it is impossible to compare to ADAS, but then actually doing so in the updated PDF. In any case, I am glad to see that comparison now exists. On that issue, however, I did not see a description of how you implemented/ran ADAS. Did you use our code? If so, did you modify it? Or did you re-implement it? All of those details need to be provided so a reader can judge if it was a fair comparison. Back to the new results, I think it would be fair and informative to point out that these results are up to some fixed compute budget and or provide performance curves over time so the reader gets a sense of the trends. Further, I think it would be most informative to say something like “MoLAS provides lots of domain knowledge and constraints that speed up search in the limited-compute regime (as is typically the case for hand-designed vs. less constrained search spaces), although it is possible in the limit of substantial compute ADAS would cross over and eventually outperform AgentSquare” (which is what I would expect). While that is in my view the most informative and fair, I understand that space is limited and you may not have space to include that or feel it is required.
> >
> > Regarding the prompt. My complaint was never that you used the prompt. It is that you used it and did not let people know that you did (especially in the larger context of not making it clear that the extremely related work you took the prompt from exists). We love people building off of the work that we publish! That is the main reason we conduct scientific research and share it with the world. But it is essential to give credit when you build off others' ideas (including code and prompts). Thus, we are not (nor were we ever) asking you to rewrite the prompt or not use it. Instead, regarding the prompt specifically, we just wanted you to make it clear to the reader that you copied and modified prompts from ADAS. You say you have done that now, which is great.
> >
> > Regarding the new research area you propose. As you write many times in the updated PDF, MoLAS is a more constrained agent search than ADAS. Since ADAS is the unconstrained version, and is a research area we pointed out is forming and gave a name to, I think it would be most informative for you to say that you are naming a new research area called MoLAS that is part of a broader research area called ADAS. I think letting the reader know that gives honest credit to the history of ideas in this space, and helps the reader properly situate MoLAS in the world where ADAS already exists.
> >
> > To my eye, your revised PDF removes your prompts. Why? I think for full transparency those should be included (in almost every paper that uses prompts). That would also be a nice place to mention you copied and modified ADAS prompts (which, again, is totally fine and actually encouraged provided you explicitly state this fact).
> >
> > Minor: I think it would make the writing make more sense, flow better, and be more appropriate to remove the word “Besides” on line 520 of the updated PDF. The word makes it seem like this is an “aside” (a not super relevant) sentence/point, but it is very relevant.
> >
> >
> > Overall, I am happy to see the reviewers fix the main issues that existed in the original manuscript. I think the above issues should be resolved (or at least addressed), but with them fixed I think the paper would constitute a nice contribution to the machine learning community, and I would be likely to change my score substantially to reflect that.
> >
> > [response to the response 2/2]

---

> > > ### Author Response · Authors · 2024-11-30
> > > **Further Response to Reviewer Dm8t (2/2)**
> > >
> > > **Response to the other points in your initial review:**
> > >
> > > **About the surrogate model performance and the number of data points in Figure 5:**
> > >
> > > We want to clarify that the result shown in Figure 5 is a standalone experiment, not extracted from the search process. That is why the number of data points does not match with Figure 4. The tested agents in Figure 5 were generated through random sampling by randomly combining existing modules. Then the surrogate model is prompted with in-context information (not zero-shot) including (1) function descriptions of four types of modules (2) module combinations of previously tested agents and their performance. These in-context examples within the modular space provides valuable performance insights of existing modules, which contributes to more accurate predictions. The code for these experiments is available at: https://www.dropbox.com/scl/fo/hcd83ax4u7vi2buqjgj03/ABYkLMM3ZdxAfNeyoJWNmls?rlkey=vck4r1cznqfyo777wn8p6rkof&st=h4pp54c0&dl=0.
> > >
> > > To further validate the consistency between these standalone experiments and the performance during the search process, we provide more experiment results of predicting the performance of the dynamically searched agents: https://www.dropbox.com/scl/fi/wjkuemg3orj7o8civp9wl/fig5-more-data-points.pdf?rlkey=70a0g9p35ena30bzfew8rracc&st=kvsh6y5d&dl=0. The predicted performance closely aligns with the evaluation performance, showing the consistent effectiveness of the proposed surrogate model. We have added these results to the Appendix.
> > >
> > > **About the usefulness of memory modules:**
> > >
> > > The usefulness of the memory module can be interpreted from Figure 4. In the left search trajectory, there's a substantial performance improvement at the 3rd iteration. The performance of the new agent equipped with Voyager memory module is 0.554, whereas the performance of the agent without memory module is only 0.479. Similarly, in the 5th iteration of the right sub-figure, the existence of the memory module from Dilu results in a great performance gap (0.537 vs. 0.591). These observations can demonstrate the memory module's effectiveness.
> > >
> > > **Regarding the missing reference:**
> > >
> > > Thanks for the suggestion. We have included a sentence at the beginning of Section 2.1 (Background) stating that using LLMs for automatic optimization has been a widely studied topic, including important applications in code generation [citing https://arxiv.org/abs/2206.08896] and neural architecture search.

---

> ### Author Response · Authors · 2024-11-30
> **Further Response to Reviewer Dm8t (1/2)**
>
> We would like to sincerely thank the reviewer for acknowledging the independent origin of our work and for recognizing the efforts we have made to address the issues raised. We also regret any misunderstanding caused by the wording in our original submission. It was never our intention to downplay the contributions of prior work, and we apologize for any confusion this may have caused. Throughout the review process, we have acted in good faith to address all concerns. We are pleased that the revisions we have made align with your suggestions and that we are now developing a mutual understanding of how to improve the paper further. We deeply appreciate this valuable exchange and believe it has significantly enhanced the quality of our work.
>
> Regarding the remaining issues, we have made further revisions and would like to provide the following response. We hope these adjustments adequately address your points.
>
>
>
> **Regarding the implementation of ADAS in our comparison:**
>
> We use the official code of ADAS and make small modifications to adapt it  to our tasks. The modifications are mainly on three aspects: (1) adjusting the task descriptions, instructions and examples in the meta prompt; (2) replacing the fitness value of each agent with its performance on our tasks; and (3) changing the evaluation procedure to align with our tasks. **We have added in these details in baseline description in Appendix A.1 to clarify these issues.** The code for implementing ADAS on our tasks is available at: https://www.dropbox.com/scl/fo/8wb1vk0x3kfqnqzku3r2r/AFD7Y-Fvbmh3exWUX90Gwt0?rlkey=fl8d7t9e9w2htndre52zvbrgu&st=ewoqk7od&dl=0.
>
> It is true the comparison between ADAS, AgentSquare, and other search-based baselines is conditioned on a fixed LLM token budget to ensure a fair comparison.  Specifically, since most of the token cost arises from agent evaluation, the fixed token budget is implemented by keeping the same number of search iterations across all methods. To inform the readers about the potential of substantial compute, **we have added a sentence in experiment result discussion** (Line 414): while in principle ADAS has the potential to discover more sophisticated agents by searching in the entire code space, it may require more iterations (and thus higher LLM token usage) to achieve this.
>
> **Regarding the research area:**
>
> We have strengthened the discussion on the search space of MoLAS and ADAS problem in the third paragraph of Introduction. We clarify that MoLAS is a guided and constraint searching problem in the modular design space, which is a subset of the entire code search proposed in ADAS. However, MoLAS has a nice feature of allowing researchers to easily recombine the strengths of prior successful designs. Besides, the discovered modules can cooperate with each other through the standard IO interfaces, facilitating broad module reuses in the research community.
>
> We have also revised the word "Besides" to "More relevantly" in Line 520, and added in the prompt to improve transparency.
>
> Please note that the above revisions are not correctly reflected in the current PDF in OpenReview, because it has turned off the pdf update option. These revisions will show up in future version of this paper, please see https://www.dropbox.com/scl/fi/28fo9nsjma79hnt4qqr5u/ICLR_2025_AgentSquare-1201.pdf?rlkey=yk40flpteqe3aiwicyvtd7v0t&st=8u0w416o&dl=0 .

---

> > ### Comment · Reviewer_Dm8t · 2024-12-03
> >
> > I thank the authors for the stance and attitude in their most recent reply. I am glad to hear you have found the exchange valuable and that you agree it has “significantly enhanced the quality of [y]our work.”
> >
> > Overall I think the authors have fixed all the main issues I raised. I am glad to see that. With all these fixes in place, which I think were necessary, I now think the paper makes a valuable contribution worthy of publication at ICLR. I have thus changed my score all the way from a 1 to a 6. I wish the authors luck with the submission. I hope you carry these lessons beyond this paper into future submissions. I also hope you continue to do research in this exciting area.

---

> > > ### Author Response · Authors · 2024-12-03
> > >
> > > Thank you for your positive feedback.

---

### Official Review · Reviewer_fhp6 · 2024-10-27

**Soundness:** 2
**Presentation:** 2
**Contribution:** 3
**Rating:** 6
**Confidence:** 3

**Summary:**

This paper proposed a framework for automatically selecting and optimizing for off the shelf LLM-based methods from existing literature. It proposes a new concept:MoLAS to enhance agent performance through modular evolution and recombination. The design includes four fundamental modules: Planning, Reasoning, Tool Use, and Memory and uses an in-context performance predictor to efficiently evaluate designs, outperforming manually crafted agents across six benchmarks by an average of 17.2%.  In sum up, AgentSquare provides a platform for researchers to reuse and extend prior successful LLM agent designs.

**Strengths:**

This method shows it's originality in using evolutionary and recombined mechanisms to automatically explore optimal combinations, effectively consolidating prior research efforts. It also provides empirical testing across diverse benchmarks, showing improvement over handcrafted and prior methods. The research extends to its broad applications and the potential to unify efforts within the LLM agent community, reducing reliance on task-specific human design part and enabling a more systematic exploration of agent architectures.

**Weaknesses:**

The paper lacks clarity regarding the definition of certain components of the method, particularly the performance evaluation function mentioned in Section 3.1. It is unclear whether this refers to the API cost introduced later or another evaluation metric. Additionally, the method shows limited novelty, as it primarily focuses on leveraging LLMs to recombine and select existing components rather than introducing a fundamentally new application or capability for LLMs. Although the proposed approach does achieve better performance than existing handcrafted methods, it does not address or eliminate the core limitations of prior works, given that it relies heavily on reusing previous designs.

**Questions:**

1. What is the evaluation and metric function to optimize the searching process ?
2. How does it scales if extending to multi-agent problem?
3. Is it able to solve manipulation tasks ?

---

> ### Author Response · Authors · 2024-11-18
> **Response to Reviewer fhp6**
>
> Thanks for your valuable feedback, we provide the following response for your concerns:
>
> **Q1: The paper lacks clarity regarding the definition of certain components of the method, particularly the performance evaluation function mentioned in Section 3.1. It is unclear whether this refers to the API cost introduced later or another evaluation metric.**
>
> **Response:** Sorry for the ambiguity. The performance evaluation function refers to the task completion accuracy of agents instead of API costs. The specific metric varies in different tasks, following the evaluation settings in their original work. Specifically, the evaluation metric is "success rate" for ALFWorld and M3ToolEval, "task score" for Webshop, "progress rate" for SciWorld and PDDL, and "micro pass rate" for TravelPlanner. We have added these explanations in Section A.1 (experimental setup) of the Appendix .
>
> **Q2: Additionally, the method shows limited novelty, as it primarily focuses on leveraging LLMs to recombine and select existing components rather than introducing a fundamentally new application or capability for LLMs. Although the proposed approach does achieve better performance than existing handcrafted methods, it does not address or eliminate the core limitations of prior works, given that it relies heavily on reusing previous designs.**
>
> **Response:**
>
> We would like to further clarify the novelty of our work and key innovation beyond prior approaches.
>
> The first innovation is a novel modular design space for LLM agents, accompanied by extensive efforts in large-scale module standardization. Compared with direct agent code or prompt search, this modular design space offers a more systematic and fruitful way to design and optimize LLM agents.
>
> The second technical innovation is an efficient search framework based on this design space, which **achieve both the reuse of existing agent designs and the discovery of innovative ones**. The search framework includes (1) module recombination, which strategically integrates existing modules to uncover novel synergies, and (2) module evolution, which leverages the in-context reasoning capabilities of LLMs to generate entirely new modules. These two operations work together in our search process, ensuring to explore far beyond the constraints of previous designs.
>
> Finally, our work can provide a community-driven platform to consolidate efforts across researchers. By standardizing modules and maintaining an open-source repository, we address the core limitation of “reinventing the wheel” in prior works. Researchers can contribute new modules and immediately benefit from our provided ecosystem, fostering collaboration and accelerating innovation in LLM agent design.
>
> **Q3: What is the evaluation and metric function to optimize the searching process ?**
>
> **Response:** The performance evaluation function refers to the task completion accuracy of agents instead of API costs. The specific metric varies in different tasks, following the evaluation settings in their original work. Specifically, the evaluation metric is "success rate" for ALFWorld and M3ToolEval, "task score" for Webshop, "progress rate" for SciWorld and PDDL, and "micro pass rate" for TravelPlanner. In the search process, we provide detailed explanation about the evaluation metric used in the targeted task to guide the search.
>
> **Q4: How does it scales if extending to multi-agent problem?**
>
> **Response:** Our framework is practical to extend to multi-agent systems. The current work provides a standardized and efficient way to construct single-agent systems, enabling the creation of multiple customized agents with different configurations. Extending to multi-agent systems would primarily require adding a standard "communication" module to facilitate interactions among agents such as flat communication and hierarchical coordination. In this way, we can leverage the same principles of module recombination and optimization to establish multi-agent systems, maintaining scalability and flexibility.
>
> **Q5: Is it able to solve manipulation tasks ?**
>
> **Response:** Yes, we have conducted experiments on two text-based embodied agent tasks, ALFWorld and SciWorld, which include manipulation tasks. The results demonstrate that agents discovered with AgentSquare outperform others. For manipulation tasks in the real physical world, the main difference with the current text-based setting lies in the form of perception and action—shifting from text-based interaction to embodied interaction. Since the core architecture of the agents remains unchanged, the agents discovered by AgentSquare are expected to achieve superior performance in such settings as well.

---

> ### Author Response · Authors · 2024-12-03
> **A Sincere Request for Further Consideration of Our Responses**
>
> Dear Reviewer fhp6,
>
> Thanks again for your valuable comments and suggestions. We have carefully addressed all the concerns mentioned and have provided a detailed point-by-point response in the rebuttal. What's more, the following revisions have been made in the paper:
>
> (1) About the clarification of evaluation metrics, we have added more details of evaluation metrics in Line 211-212 of Section 3.1 and Line 775-778 of Section A.1 in the Appendix.
>
> (2) About the novelty, we have added more discussion of our main difference and key contribution in Line 138-144 of Section 2.1 Background.
>
> We hope the revisions and clarifications we have made adequately address your concerns.
>
> As the discussion phase is coming to a close, we kindly ask if you could take a moment to review our responses. We understand the time constraints, but we sincerely hope that the revisions address your concerns and will be reflected in the final scoring. We deeply appreciate your time and efforts in helping improve our work.
>
> Best regards,
>
> Authors

---

### Official Review · Reviewer_T4be · 2024-10-29

**Soundness:** 3
**Presentation:** 3
**Contribution:** 3
**Rating:** 6
**Confidence:** 4

**Summary:**

This paper introduces a modular design space for LLM agents named “Modularized LLM Agent Search (MoLAS)” and proposes an evolutionary framework named “AgentSquare” for optimizing and recombining the various modules (covering planning, reasoning, tool use and memory) within this design space. It further introduces a performance prediction model for ruling out unpromising candidate solutions, hence rendering the search process more efficient. Through comprehensive evaluations it is demonstrated that their autonomous, modular design of LLM agents significantly outperforms manual designs and other search algorithms, and offers interpretable insights that complement human knowledge. This paper thus contributes to more standardized and scalable development of agentic systems that exploit prior successful experience, minimizing the reliance on human intervention.

**Strengths:**

1.The paper is well-organized and uses language that is easy to understand.

2.The paper is well motivated, addressing an interesting research question. It innovatively consolidates existing (and potentially upcoming) LLM agent designs into a unified framework, and effectively leverages their successful experience for better design.

3.The experiments are conducted with both quantitative evaluations (including task performances, API costs and search trajectories), and qualitative analyses such as the specific module combinations of optimized agents and insights drawn from the newly discovered modules.

4.The authors have made their code repository available on GitHub, which ensures reproducibility and promotes future work.

**Weaknesses:**

1.There are some inconsistencies within the paper that might cause confusion. For example, Equation (2) and (3) indicate that both module recombination and evolution take past experience as input, which is not the case in Figure 3. Besides, the random initialization (as mentioned in experimental setup) seems to contradict the arguments made at the beginning of Section 3.3.

2. It seems more fair to also take into account the API cost incurred by search when you compare the performance-cost trade-offs of AgentSquare and manual design. However, the paper did not mention this information.

3.The experimental results appear to be obtained from a single run. The authors are suggested to carry out repeated experiments, and report averaged results and standard deviations to demonstrate the robustness of their approach.

**Questions:**

1.Are there any specific criteria for selecting the 16 LLM agents when constructing module pools? Besides, human labor involved in standardizing these modules might become prohibitive if one hopes to leverage a larger number of existing agents. Are there any measures to tackle this problem?

2.Could you elaborate a bit more on the evaluation of candidate solutions? Specifically, it is currently unclear when solutions are evaluated by the performance predictor and when they are tested in the real task environment.

Thank you!

---

> ### Author Response · Authors · 2024-11-18
> **Response to Reviewer T4be (Part 1)**
>
> Thanks for your valuable feedback, we provide the following response for your concerns:
>
> **Q1: There are some inconsistencies within the paper that might cause confusion. For example, Equation (2) and (3) indicate that both module recombination and evolution take past experience as input, which is not the case in Figure 3.**
>
> **Response:** Thanks for pointing out. It's right that both module recombination and evolution take past experience as input. We have corrected the illustration in Figure 3 and updated the paper.
>
> **Q2: Besides, the random initialization (as mentioned in experimental setup) seems to contradict the arguments made at the beginning of Section 3.3.**
>
> **Response:**
> Sorry for the caused misunderstanding. The two arguments refer to the initialization of different components.
>
> In Section 3.3, we describe the initialization of the module pool and performance experience pool, which are not randomized but based on prior successful agents and their performance data.
>
> In contrast, the random initialization mentioned in the experimental setup refers specifically to the initial agent configuration, which is randomly composed of modules from the pool. This random combination serves as the starting point for the search process.
>
> Thus, these two arguments are not contradictory.
>
> **Q3: It seems more fair to also take into account the API cost incurred by search when you compare the performance-cost trade-offs of AgentSquare and manual design. However, the paper did not mention this information.**
>
> **Response:** Thank you for the suggestion. We have provided the API costs incurred during the search process with AgentSquare in Table A.6 of the Appendix. However, since the search cost is a one-time expense and the discovered modules can be shared within the community, these costs can be effectively amortized. As a result, we exclude the search cost from the performance-cost trade-off analysis presented in Figures A.7–A.12. These figures focus on demonstrating that agents discovered by AgentSquare achieves the best performance-cost trade-off compared to other agents. The above discussion has been added to the paper.
>
> **Q4: The experimental results appear to be obtained from a single run. The authors are suggested to carry out repeated experiments, and report averaged results and standard deviations to demonstrate the robustness of their approach.**
>
> **Response:** Thanks for the suggestion. We have carried out repeated experiments of our method and the best baseline for each task. Each method is runned three times independently. The averaged results and standard deviations are reported as follows:
>
> *Performance comparison of searched agents from AgentSquare and the best baseline based on GPT-4o across different tasks:*
>
> | **Method**\Task | **Webshop**   | **ALFWorld**  | **SciWorld**  | **M3Tool**    | **TravelPlanner** | **PDDL**      |
> | --------------- | ------------- | ------------- | ------------- | ------------- | ----------------- | ------------- |
> | AgentSquare     | 0.601(±0.012) | 0.706(±0.008) | 0.799(±0.019) | 0.520(±0.006) | 0.577(±0.005)     | 0.666(±0.002) |
> | Best Baseline   | 0.545(±0.007) | 0.632(±0.003) | 0.771(±0.004) | 0.501(±0.010) | 0.559(±0.006)     | 0.652(±0.006) |
>
>
> *Performance comparison of searched agents from AgentSquare and the best baseline based on GPT-3.5 across different tasks:*
>
> | **Method**\Task | **Webshop**   | **ALFWorld**  | **SciWorld**  | **M3Tool**    | **TravelPlanner** | **PDDL**      |
> | --------------- | ------------- | ------------- | ------------- | ------------- | ----------------- | ------------- |
> | AgentSquare     | 0.617(±0.007) | 0.643(±0.007) | 0.438(±0.011) | 0.286(±0.005) | 0.518(±0.003)     | 0.216(±0.008) |
> | Best Baseline   | 0.583(±0.008) | 0.598(±0.009) | 0.360(±0.025) | 0.233(±0.001) | 0.494(±0.019)     | 0.211(±0.009) |
>
> From the result, it can be observed that AgentSquare consistently outperforms the best baseline for all tasks, demonstrating the superiority of our approach.

---

> ### Author Response · Authors · 2024-11-18
> **Response to Reviewer T4be (Part 2)**
>
> **Q5: Are there any specific criteria for selecting the 16 LLM agents when constructing module pools? Besides, human labor involved in standardizing these modules might become prohibitive if one hopes to leverage a larger number of existing agents. Are there any measures to tackle this problem?**
>
> **Response:**
>
> **Selection criteria:** We perform a comprehensive literature review of publications from NeurIPS, ICML, and ICLR over the past three years. The review focuses on papers with the keywords “LLM”, “Agent”, or “Large Language Model” in their titles while excluding works related to multi-agent systems or agents that require additional training. Note that our aim is not to propose the most comprehensive, one-for-all LLM agent design space, but to offer a standardized framework that enables the recombination of existing agents and facilitates the discovery of new ones (as described in Section 2 of the paper). As a result, we sort out 16 popular LLM agents as the current version. Importantly, this module pool can be easily extended to new agents, with our standardized IO interfaces of agent modules.
>
> **Solution about the effort of standardizing agent modules:** We have open-sourced all standardized modules and maintain a leaderboard to encourage contributions from the research community. Our platform allows researchers to design new modules in a standardized way, enabling easy integration into the framework and the rapid development of new agents. By leveraging community crowdsourcing, we aim to collectively solve this problem and advance LLM agent development.
>
>
>
> **Q6: Could you elaborate a bit more on the evaluation of candidate solutions? Specifically, it is currently unclear when solutions are evaluated by the performance predictor and when they are tested in the real task environment.**
>
> **Response:** Thanks for the suggestion. We clarify that the performance predictor is applied during the module recombination stage, not during the module evolution stage. Specifically, the experience pool is initialized with real-tested performance of existing agents we sorted out.
>
> During the module evolution process, newly created modules are tested directly in the real task environment because these modules never appear in the experience pool. In this situation, it would be unreasonable to utilize the performance predictor for accurate predictions, as it works via in-context reasoning based on historical performance of agents with overlapping modules.
>
> During the module recombination stage, the newly proposed agents are evaluated by the performance predictor, which leverages in-context reasoning based on past agent combination performance to provide efficient performance prediction. We have refined the descriptions in Sections 3.6 to make it clearer.

---

> > ### Comment · Reviewer_T4be · 2024-11-26
> > **Response to authors**
> >
> > Thank you for the reply.

---

### Official Review · Reviewer_funC · 2024-11-04

**Soundness:** 3
**Presentation:** 3
**Contribution:** 3
**Rating:** 6
**Confidence:** 3

**Summary:**

The paper introduces AgentSquare, a framework designed to automatically optimize LLM agent architectures within a modular design space. It proposes a novel approach, termed Modularized LLM Agent Search (MoLAS), by defining a modular architecture divided into Planning, Reasoning, Tool Use, and Memory modules. AgentSquare employs module evolution and recombination mechanisms, along with a performance predictor, to explore and identify optimal combinations within the design space efficiently. The framework is evaluated on six benchmarks, showing a 17.2% performance improvement over existing hand-crafted agent designs.

**Strengths:**

- Modularity and Reusability: The modular allows the reuse and recombination of components, which aligns well with LLM advancements in modularization and scalability.

- Effective Search Mechanism: The combination of module evolution, recombination, and performance prediction seems to be a robust optimization strategy. The proposed performance predictor effectively reduces evaluation costs, addressing practical limitations in real-world deployments of LLM agents.

- Comprehensive Evaluation: Benchmarks across domains such as web applications, embodied AI, and gaming provide evidence of AgentSquare’s efficacy and generalizability.

- Interpretable Insights: The framework’s ability to provide human-interpretable design insights is a useful addition, potentially helping in  aiding the design and tuning of future LLM-based agents.

**Weaknesses:**

My main issue with the paper is that while the modular approach is beneficial in the short-term, it may limit flexibility by enforcing predefined components. Extending the modular design to allow more dynamic, task-specific modules could enhance its applicability.

Additionally, the framework’s reliance on LLM-driven suggestions for module evolution and recombination could inherit biases or inefficiencies from the LLM models themselves, potentially limiting the quality of novel configurations.

Minor comment:
- Many citations miss parenthesis around them. Try using \citep instead.

**Questions:**

The authors write that: “In contrast, module-level searching methods including random and Bayesian search lack a clear and insightful search direction." Where do we see this? When looking at the figures, it seems that random search is a pretty competitive baseline?

---

> ### Author Response · Authors · 2024-11-18
> **Response to Reviewer funC (Part 1)**
>
> Thanks for your valuable feedback, we provide the following response for your concerns:
>
> **Q1: My main issue with the paper is that while the modular approach is beneficial in the short-term, it may limit flexibility by enforcing predefined components. Extending the modular design to allow more dynamic, task-specific modules could enhance its applicability.**
>
> **Response:**
> We would like to clarify that although we define standard module interfaces for LLM agents, the agent modules are not constrained to predefined designs. In fact, our method has adaptively discovered many innovative module designs via module evolution.  Our work aims to **strike a balance between framework generality and task-specific flexibility**. This is similar to the modern computer architectures, which may not be optimized for specific tasks but can be widely applied to various scenarios due to their modularity. Beyond performance benefits, our modular framework also provides long-term advantages by reducing redundant efforts and fostering community collaboration through a shared, extensible design space.
>
> **About the generality of our framework:** Our framework primarily models the agent’s cognitive process, aligned well with (1) existing successful agent designs and (2) widely accepted perspectives on key components of LLM agents, such as Lilian Weng's views on agent systems [1] and Andrew Ng's insights on AI agentic workflow [2].  Extensive experimental results on 16 existing agent designs across 6 representative task scenarios also validates the modular framework's wide applicability.
>
> **About the flexibility of our framework:** The implementation within each module is actually highly flexible, allowing task-specific adaptations. For instance, the tool-use module in our framework can be tailored to plug-and-play tool sets related to the specific task (e.g., "fight search" and "restaurant search" tools for travel planning tasks,  "click_url" and "go_to_previous_page" tools for web-browsing tasks).  This is analogous to external devices in modern computers, which can be conveniently connected and replaced to handle different tasks. Such inherent flexibility ensures the framework's applicability to a wide range of dynamic, task-specific scenarios.
>
> We will add more discussion about the balance between flexibility and generality of the modular design and the possible future direction in the revised paper.
>
> [1] https://lilianweng.github.io/posts/2023-06-23-agent/
>
> [2] https://x.com/AndrewYNg/status/1770897666702233815
>
> **Q2: Additionally, the framework’s reliance on LLM-driven suggestions for module evolution and recombination could inherit biases or inefficiencies from the LLM models themselves, potentially limiting the quality of novel configurations.**
>
> **Response:** LLMs are demonstrated to possess powerful in-context reasoning capabilities, making them effective for heuristic search problems. Utilizing LLMs for evolutionary search has been widely applied to the neural architecture search (NAS) area [1] [2] [3]. To mitigate LLM's inherit biases, we provide LLMs with rich in-context information including specific task descriptions, prior successful agent module designs, performance of tested agent configurations, etc. These in-context knowledge is iteratively refined and expanded as more novel agents are discovered, further enhancing the search process.
>
> More importantly, the proposed modular framework also supports classical search methods without LLMs, such as random and Bayesian search, which we have tested in our experiments. The results in Table 1 show that LLM-driven search achieves better performance, verifying the quality and effectiveness of the LLM-based approach. In addition, our ablation studies (Table 2) further demonstrate the effectiveness of LLM-driven module recombination and evolution, showing a clear performance contribution.
>
> [1] Chen, A., Dohan, D., & So, D. (2024). EvoPrompting: language models for code-level neural architecture search. *Advances in Neural Information Processing Systems*, *36*.
>
> [2] Jawahar, Ganesh, et al. "LLM Performance Predictors are good initializers for Architecture Search." *arXiv preprint arXiv:2310.16712* (2023).
>
> [3] Nasir, Muhammad Umair, et al. "LLMatic: neural architecture search via large language models and quality diversity optimization." *Proceedings of the Genetic and Evolutionary Computation Conference*. 2024.

---

> ### Author Response · Authors · 2024-11-18
> **Response to Reviewer funC (Part 2)**
>
> **Q3: Minor comment: Many citations miss parenthesis around them. Try using \citep instead.**
>
> **Response:** Thanks for pointing out that, we have corrected this issue and updated the paper.
>
> **Q4: The authors write that: “In contrast, module-level searching methods including random and Bayesian search lack a clear and insightful search direction." Where do we see this? When looking at the figures, it seems that random search is a pretty competitive baseline?**
>
> **Response:** These two module search methods indeed have competitive performance compared with prompt search methods, because they can reuse existing successful modules to easily form a powerful agent. This fact indicates the importance of searching in the modular design space.
>
> However,  as shown in Figure 4, while random and Bayesian search occasionally find competitive solutions, it demonstrates a "flatter" search trajectory compared to AgentSquare. For example, in the left panel of Figure 4, the 7-iteration AgentSquare search has outperformed the 15-iteration random and Bayesian search. In the right panel of Figure 4, the 5-iteration AgentSquare search has outperformed the 18-iteration random and Bayesian search. These observations reflect that AgentSquare provides a clearer search direction and achieves consistent improvement of agents. Moreover, a key advantage of AgentSquare search is its ability to discover new modules, whereas random and Bayesian search are restricted to existing modules. We will add more discussion about this in the revised paper.

---

> > ### Comment · Reviewer_funC · 2024-11-26
> >
> > Thank you for the clarifications. I noticed that e.g. Figure 4 doesn't show standard deviation for the different methods. Across how many runs did you average these results? If it's just one run, the different approaches could have just been lucky?

---

> > > ### Author Response · Authors · 2024-11-30
> > > **Further Response to Reviewer funC**
> > >
> > > Thanks for your further feedback. To strengthen this experiment, we have carried out repeated experiments of our method and the best baseline for each task. Each method is run three times independently. The averaged results and standard deviations are reported as follows. The paired t-test analysis shows the performance improvement of AgentSquare is statistically significant in all tasks (p-value<0.05). Please find the detailed experiment result in below table.
> > >
> > > *Performance comparison of searched agents from AgentSquare and the best baseline based on GPT-4o across different tasks:*
> > >
> > > | **Method**\Task | **Webshop**   | **ALFWorld**  | **SciWorld**  | **M3Tool**    | **TravelPlanner** | **PDDL**      |
> > > | --------------- | ------------- | ------------- | ------------- | ------------- | ----------------- | ------------- |
> > > | AgentSquare     | 0.601(±0.012) | 0.706(±0.008) | 0.799(±0.019) | 0.520(±0.006) | 0.577(±0.005)     | 0.666(±0.002) |
> > > | Best Baseline   | 0.545(±0.007) | 0.632(±0.003) | 0.771(±0.004) | 0.501(±0.010) | 0.559(±0.006)     | 0.652(±0.006) |
> > > | p-value         | 0.0000        | 0.0000        | 0.0000        | 0.0338        | 0.0002            | 0.0143        |
> > >
> > > We will add standard deviation to Figure 4 in the revised paper.

---

> ### Author Response · Authors · 2024-12-03
> **A Sincere Request for Further Consideration of Our Responses**
>
> Dear Reviewer funC,
>
> Thank you once again for your valuable feedback during the review process. We have carefully addressed all the points raised and have provided a detailed point-by-point response in the rebuttal. Moreover, we have made the following revisions in the paper:
>
> (1) About the balance between flexibility and generality of our method, we have added discussion of our balance of reusing existing designs and discovering innovative designs in Line 142-144 of Section 2.1 Background.
>
> (2) About citation formatting, we have corrected the issue throughout the paper.
>
> (3) About the reliability of LLM-driven agent design, we have supplemented more relevant works to prove that in Line 138-141 of Section 2.1 Background.
>
> We sincerely hope that the revisions and clarifications we've made meet your expectations.
>
> As the discussion phase is coming to a close, we kindly ask if you could take a moment to review our responses. We understand the time constraints, but we are hopeful that the revisions address your concerns and will be reflected in the final scoring. We truly appreciate your consideration. Thank you for your time and support in refining our work.
>
> Best regards,
>
> Authors

---

> > ### Comment · Reviewer_funC · 2024-12-03
> >
> > Thank you for your response, which clarified my remaining questions. Based on those, I'll increase my score.

---

### Meta-Review · Area_Chair_MmmK · 2024-12-23

**Metareview:**

Wow, working through the reviewer discussion for this paper was a trip. As someone who has seen multiple cases of plagiarism accusations, both well-founded and unfounded, it was really interesting to see how this played out. As it turns out, I am happy with the outcome. It seems clear to me that the research described in this paper was independently conceived and executed, but also that the original manuscript was downplaying the role of the ADAS paper (which was published to arXiv earlier) to the point of seemingly trying to obscure it. The new version of the AgentSquare paper is much improved in that regard.

The AgentSquare paper is also a good paper overall - an interesting new approach to an important problem with good results. Clear accept.

**Additional Comments On Reviewer Discussion:**

"Spicy"

---

### Decision · Program_Chairs · 2025-01-22

Accept (Poster)